# Manganese is a potent inducer of lysosomal activity that inhibits *de novo* HBV infection

Lin Yu[1,2ↄ], Hao Chang[1,2ↄ], Wentao Xie[1,2,3], Yuan Zheng[1,2], Le Yang[1,2], Qiong Wu[1,2], Fan Bu[1,2], Yuanfei Zhu[1,2], Youhua Xie[1,2], Guoyu Pan[4]*, Ke Lan[5]*, Qiang Deng [1,2]*

**1** Key Laboratory of Medical Molecular Virology (MOE/NHC/CAMS), Shanghai Institute of Infectious Disease and Biosecurity, School of Basic Medical Sciences, Fudan University, Shanghai, China, **2** Shanghai Frontiers Science Center of Pathogenic Microorganisms and Infection, Fudan University, Shanghai, China, **3** Department of Infectious Diseases, Shanghai Key Laboratory of Infectious Diseases and Biosafety Emergency Response, National Medical Center for Infectious Diseases, Huashan Hospital, Shanghai, China, **4** Shanghai Institute of Materia Medica, Chinese Academy of Sciences, Shanghai, China, **5** State Key Laboratory of Virology, College of Life Sciences, Wuhan University, Wuhan, China

ↄ These authors contributed equally to this work.
* gypan@simm.ac.cn (Gy P); klan@whu.edu.cn (KL); qdeng@fudan.edu.cn (QD)

**Data Availability Statement:** All relevant data are within the manuscript and its Supporting information files.

**Funding:** This work was supported by the Shanghai Municipal Science and Technology Major

## Abstract

Sodium taurocholate co-transporting polypeptide (NTCP) has been identified as an entry receptor for hepatitis B virus (HBV), but the molecular events of the viral post-endocytosis steps remain obscure. In this study, we discovered that manganese (Mn) could strongly inhibit HBV infection in NTCP-reconstituted HepG2 cells without affecting viral replication. We therefore profiled the antiviral effects of $Mn^{2+}$ in an attempt to elucidate the regulatory mechanisms involved in early HBV infection. Intriguingly, $Mn^{2+}$ conspicuously stimulated lysosomal activity, as evidenced by hyperactivation of mTORC1 and increased endo/lysosomal acidity. After HBV-triggered internalization, the NTCP receptor was sorted to late endosomal compartments by the ESCRT machinery in concert with the invading virion. The establishment of HBV infection was found to be independent of lysosomal fusion-driven late endosome maturation; $Mn^{2+}$-induced lysosomal hyperfunction virtually impaired infection, suggesting that virions may gain cytosolic access directly from late endosomes. In contrast, suppression of lysosomal activity substantially enhanced HBV infection. Prolonged mTORC1 inactivation facilitated viral infection by depleting lysosomes and accelerating endocytic transport of virions. Notably, treatment with the natural steroidal alkaloid tomatidine recapitulated the effects of $Mn^{2+}$ in stimulating lysosomal activity and exhibited potent anti-HBV activity in HepG2-NTCP cells and in proliferating human hepatocyte organoids. These findings provide new insights into the post-endocytosis events of HBV infection. The negative regulation of early HBV infection by endo/lysosomal activity makes it a promising target for antiviral therapies.

## Author summary

Despite the identification of NTCP as a functional HBV receptor over a decade ago, the later stages of the HBV entry process remain poorly understood. Inspired by a previous

Project (ZD2021CY001 to QD), the National Natural Science Foundation grants of China (82072279 to QD; 82372233 to QD; 81871647 to QD) and the Innovation Fund for Medical Sciences from Chinese Academy of Medical Sciences (2019-I2M-5-040 to QD). The funders had no role in study design, data collection and analysis, decision to publish, or preparation of the manuscript.

**Competing interests:** The authors have declared that no competing interests exist.

publication, we tested the possible effect of manganese (Mn) on HBV infectivity. To our surprise, although $Mn^{2+}$ did not induce a type I interferon response, it was still able to inhibit HBV infectivity in HepG2-NTCP cells. Using loss-of-function assays with pharmacological inhibitors and RNA interference, we found that after internalization, HBV and the NTCP receptor are co-sorted via the ESCRT machinery into the late endosome, where the virions achieve endosomal escape. Notably, endosome-lysosome fusion is not required for HBV infection. In contrast, lysosomal hyperfunction, induced by $Mn^{2+}$ or the natural steroidal alkaloid tomatidine, impairs *de novo* HBV infection, while the use of lysosomal inhibitors facilitates this process. The suppressive impact of lysosomal activity on the early HBV infection renders it a promising target for antiviral therapies.

## Introduction

The endosomal system is composed of interconnected membrane-bound organelles, including early endosomes, multivesicular bodies (MVB)/late endosomes, and endolysosomes (late endosome–lysosome hybrids) that undergo a dynamic maturation process through fission and fusion cycles [1,2]. As the endosomes mature, the lumen becomes increasingly acidic, with endolysosomes being the principal intracellular sites of acid hydrolase activity [3,4]. Primary lysosomes are cathepsin-inactive and non-acidic storage organelles. Small GTPase RAB7-mediated fusion of lysosomes with late endosomes or autophagosomes leads to the formation of highly acidic and degradative lysosomal vacuoles [5].

Hepatitis B virus (HBV) infection is a global public health concern. Sodium taurocholate co-transporting polypeptide (NTCP) has been identified as a functional receptor with high affinity for HBV [6,7]. There is increasing evidence that HBV is internalized into hepatocytes via NTCP-mediated endocytosis, although the molecular events involved in the post-endocytosis steps remain obscure [8,9]. In HBV-permissive differentiated HepaRG cells, HBV infection is strongly dependent on the expression of RAB5 and RAB7 [10]. RAB5 is involved in the maturation of the early endosome into the multivesicular late endosome, and at the late endosome, RAB5 is replaced by RAB7 which then interacts with the HOPS complex to promote the fusion of the mature late endosome with the lysosome. Despite the importance of RAB7, the lysosomal compartment is speculated to be a dead-end branch for early HBV infection, primarily because early HBV (or duck HBV) infection does not rely on low pH-dependent proteolytic activity [10–12]. However, additional experimental evidence is required to validate this hypothesis.

The mechanistic target of rapamycin complex 1 (mTORC1) is a master regulator of cell growth and metabolism [13]. Nutrients including amino acids, glucose and lipids drive the physical localization of mTORC1 to endo/lysosomal membranes, where the kinase activity of mTORC1 is turned on by GTP-bound RHEB, which is differentially regulated by growth factor signaling. The heterodimeric Rag GTPases, Ragulator complex, amino acid transporter SLC38A9, and vacuolar adenosine triphosphatase (v-ATPase) form a signaling system that is necessary for the activation of mTORC1 by amino acids. V-ATPase is primarily responsible for acidification throughout the endocytic pathway, by pumping protons into the interior of organelles coupled with ATP hydrolysis [14]. Notably, the v-ATPase is also a positive regulator of the mTORC1 signaling pathway, presumably by stimulating the guanine nucleotide exchange factor (GEF) activity of Ragulator [15,16].

Manganese (Mn) is an essential metal involved in a number of fundamental cellular processes. Despite its importance, excessive $Mn^{2+}$ accumulation is toxic, especially to the central

nervous system, and results in neurodegeneration and related diseases [17,18]. Cytosolic $Mn^{2+}$ has recently reported to be required for host defense against DNA viruses by increasing the sensitivity of the DNA sensor cGAS and its downstream adaptor protein STING [19]. Intrigued by this finding, we examined whether $Mn^{2+}$ possesses antiviral activity against HBV which is classified as an atypical DNA virus. Interestingly, $Mn^{2+}$ was shown to be unable to induce interferon-stimulated gene (ISG) expression in HepG2 cells. Nevertheless, the treatment with $MnCl_2$ markedly inhibited NTCP-mediated *de novo* HBV infection without affecting intracellular viral replication.

In-depth mechanistic analyses in this study revealed that administration of $MnCl_2$ resulted in hyperactivation of mTORC1, and the endolysosomal compartments in $MnCl_2$-treated cells exhibited a highly acidic pH and increased degradative capacity. Our study also revealed that the HBV virion, along with the NTCP receptor, is trafficked to the late endosome via the endosomal sorting complex required for transport (ESCRT) machinery [20]. Since lysosomal fusion is not required for establishing infection, it is believed that HBV particles access the cytoplasm through late endosomal compartments, potentially via fusion with the endosomal limiting membrane. Unlike many other enveloped viruses that employ low pH to facilitate membrane fusion [21], raising endolysosomal pH or inhibiting endosome-lysosome fusion promoted HBV infection. Additionally, we found that mTORC1-regulated endocytic trafficking and lysosomal function profoundly influenced the fate of invading viral particles.

$Mn^{2+}$ is not druggable because it can cause neurotoxicity depending on the amount and duration of exposure. Tomatidine is a natural steroidal alkaloid that is abundant in unripe green tomatoes [22]. We showed that tomatidine had potent anti-HBV activity and recapitulated the effects of $Mn^{2+}$ in stimulating mTORC1 hyperactivation and endo/lysosomal acidification. Our data provide insight into understanding the post-endocytosis step of *de novo* HBV infection, which may represent a pivotal target for the treatment of HBV infection.

## Results

### $MnCl_2$ treatment inhibited *de novo* HBV infection without impacting viral replication

Cytosolic $Mn^{2+}$ is a potent activator of the cGAS-STING pathway which is required for host defense against DNA viruses [19]. Murine and human hepatocytes are thought to lack a functional DNA-sensing pathway because they do not express the key adaptor protein, STING [23,24]. There is a low-level of expression of this protein in the commonly used hepatoma cell line HepG2 (**S1A Fig**). However, treatment of HepG2 cells with $MnCl_2$, at doses up to 100 μM, did not result in significant interferon (IFN)-β production (**S1B Fig**). Compared to the cGAS product cGAMP, $MnCl_2$ treatment did not significantly increase IFN-stimulated gene (ISG) expression, including that of *Mx1*, *ISG15*, and *OAS1* (**S1C Fig**).

As indicated above, $Mn^{2+}$ appears to be incapable of stimulating type I IFN responses in HepG2 cells. However, by evaluating the expression of supernant HBeAg and the intracellular HBV core protein (HBc), it was found that treatment with $MnCl_2$ at a concentration of 10 μM (**S2A Fig**) or 40 μM (**Fig 1A**) markedly inhibited HBV infection in HepG2 cells reconstituted with NTCP without affecting cell viability (**Fig 1B**). To better understand the antiviral effects of $Mn^{2+}$, we performed a time-of-addition study with $MnCl_2$ added to HepG2-NTCP cells pre-, during-, pre- and during-, and post-infection, respectively (**Fig 1C**). Notably, pretreatment with $MnCl_2$ demonstrated a markedly enhanced anti-HBV effect compared to cells receiving $MnCl_2$ at 12 h post-infection, suggesting that $Mn^{2+}$ may primarily act at an early stage before the establishment of HBV infection. Corroborating this finding, the administration of $MnCl_2$ did not influence the viral replication, or the secretion of HBeAg in HepG2 cells

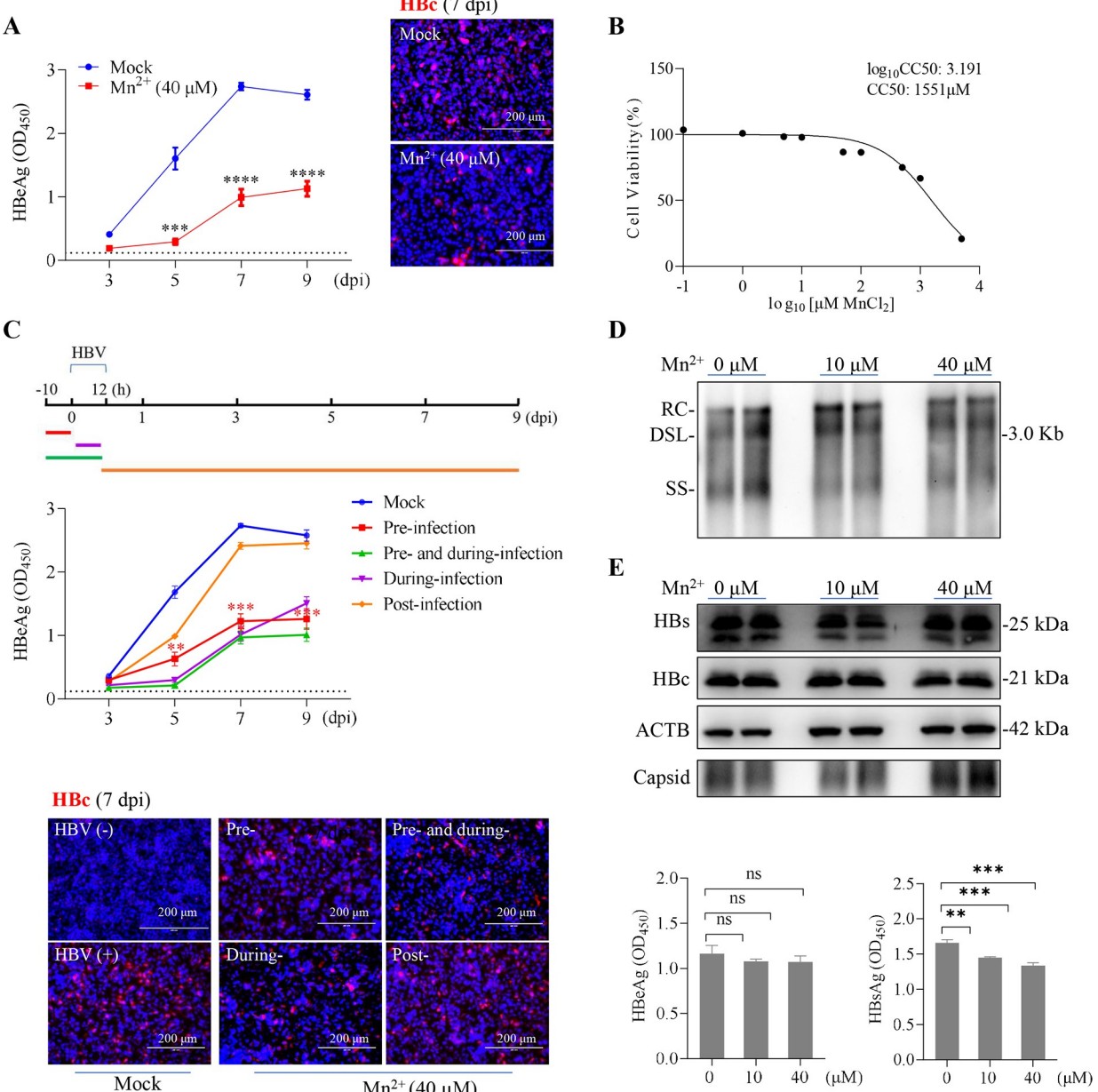

**Fig 1. Inhibition of *de novo* HBV infection by MnCl₂ in HepG2-NTCP cells.** (**A**) HepG2-NTCP cells were treated with or without MnCl₂ (40 µM) for 10 h, followed by an additional 12 h incubation with HBV inoculum (see Materials and Methods). Left, secreted HBeAg was determined at the indicated time points (*n* = 3). The dotted line indicates the cutoff value. Right, representative images of immunofluorescence labeling of intracellular HBc (red) at 7 days post-inoculation (dpi). Cell nuclei were stained with DAPI (blue). (**B**) Dose-response curve showing the cytotoxicity of MnCl₂ in HepG2-NTCP cells determined by the CCK8 assay performed in triplicate. The CC50 value was calculated with GraphPad Prism software. (**C**) Time-of-addition assay of the effect of Mn²⁺ (40 µM) on *de novo* HBV infection of HepG2-NTCP cells. Upper, the experimental scheme. Middle, supernatant HBeAg levels at the indicated time points were plotted. Comparisons were made between pre-infection and post-infection treatment. Lower, immunofluorescence staining of HBc as described in (**A**). (**D and E**) HepG2 cells were transfected with a plasmid pHBV1.3 carrying a 1.3-mer overlength HBV genome. After 6 h, the cells were treated with different concentrations of MnCl₂ for further 3 days. Capsid-associated HBV DNA species were assessed by Southern blotting, with two replicates at each concentration, representing three independent experiments. RC, relaxed circular DNA; DSL, double-stranded linear DNA; SS, single-stranded linear DNA (**D**). Viral protein expression and secretion were assessed by immunoblotting (two replicates at each concentration) and ELISA (*n* = 3), respectively (**E**). Error bars indicate mean ± SD. **, $P < 0.01$; ***, $P < 0.001$; ****, $P < 0.0001$; ns, not significant.

transfected with HBV-expressing plasmids (**Fig 1D and 1E**) or in HepAD38 cells (tet-off) integrated with the HBV genome (**S2B–S2D Fig**). Treatment also had no effect on the expression of the HBc protein or the assembly of capsids. It is noteworthy that $MnCl_2$ treatment resulted in a modest reduction in HBsAg secretion, although intracellular protein levels remained unchanged (**Figs 1E and S2D**). Therefore, in subsequent studies, we used HBeAg as the principal marker to evaluate *de novo* HBV infection.

Unlike HBsAg subviral particles, which are transported to the ER-Golgi intermediate compartment and released via the conventional secretory pathway, the secretion of HBV virions relies on the ESCRT/MVB pathway [25–27]. We therefore examined virion egress in HBV-transfected Huh7 cells treated with $MnCl_2$. However, the treatment did not significantly affect the levels of virions, naked capsids, or viral DNA in the culture supernatant (**S2E Fig**).

## $Mn^{2+}$ hyperactivated mTORC1 in cells under nutrient-rich conditions

In HepG2-NTCP cells, the administration of $MnCl_2$ did not result in a discernible alteration in the overall level or cellular distribution of NTCP (**S3 Fig**). Thus, the anti-HBV effect of $Mn^{2+}$ may occur at one or several stages of HBV infection following NTCP binding, yet prior to viral RNA transcription. To explore the underlying mechanism, we performed a comparative RNA-Seq analysis on HepG2-NTCP cells treated with or without $MnCl_2$. KEGG pathway analysis revealed that differentially expressed genes (DEGs) were primarily enriched in signaling pathways related to nutrient metabolism, such as glycolysis/gluconeogenesis, butanoate metabolism, tyrosine metabolism, and the insulin signaling pathway (**Fig 2A**). A literature review suggested that many of the DEGs identified were involved in the regulation of mTORC1 activity (**Fig 2B**).

Thus, we performed a time-course experiment to characterize the effects of $Mn^{2+}$ on mTORC1 activation. In nutrient-rich culture medium, HepG2-NTCP cells exhibited constitutive activation of mTORC1, as indicated by the corresponding phosphorylation of mTORC1 effectors S6 kinase (p-S6K) and the eukaryotic translation initiation factor 4E binding protein 1 (p-4E-BP1). Interestingly, after $MnCl_2$ treatment, there was a rapid increase in p-S6K levels, which peaked after 1–2 h. The increase in p-4E-BP1 lagged behind that of p-S6K, but the duration was relatively long (up to 8 h) (**Fig 2C**, upper panels).

Activation of mTORC1 involves spatio-temporally coordinated events. The mTORC1 complex must localize to the endo/lysosomal membrane via an amino acid-dependent process, where its kinase activity is differentially stimulated by growth factor signaling [13]. In our study, we found that serum starvation had little effect on $Mn^{2+}$-stimulated mTORC1 hyperactivation, whereas this effect was completely abolished by amino acid deprivation (**Fig 2D**). Additionally, mTORC1 inhibitors Torin1 (**Fig 2E**), which targets the enzyme's catalytic center, almost fully suppressed mTORC1 activation even in the presence of $MnCl_2$ treatment. We therefore speculated that $Mn^{2+}$ might function downstream of a (serum) growth factor signaling pathway. In support of this assumption, robust AKT phosphorylation (p-Ser473-AKT) was observed in $MnCl_2$-treated HepG2-NTCP cells with a time-dependent profile comparable to that of p-S6K (**Fig 2C**, lower panels). The KEGG analysis (**Fig 2A**) showed significant activation of the MAPK pathway in $Mn^{2+}$-treated cells. This leads to the speculation that $Mn^{2+}$-induced cellular stress and MAPK activation may hyperactivate endo/lysosomal mTORC1 through intricate crosstalk between the Ras/MAPK and PI3K/AKT signaling pathways [28, 29]. Collectively, these data suggest that $Mn^{2+}$ treatment transiently hyperactivates mTORC1 signaling under nutrient-rich conditions, presumably in part via the AKT–mTORC1 pathway.

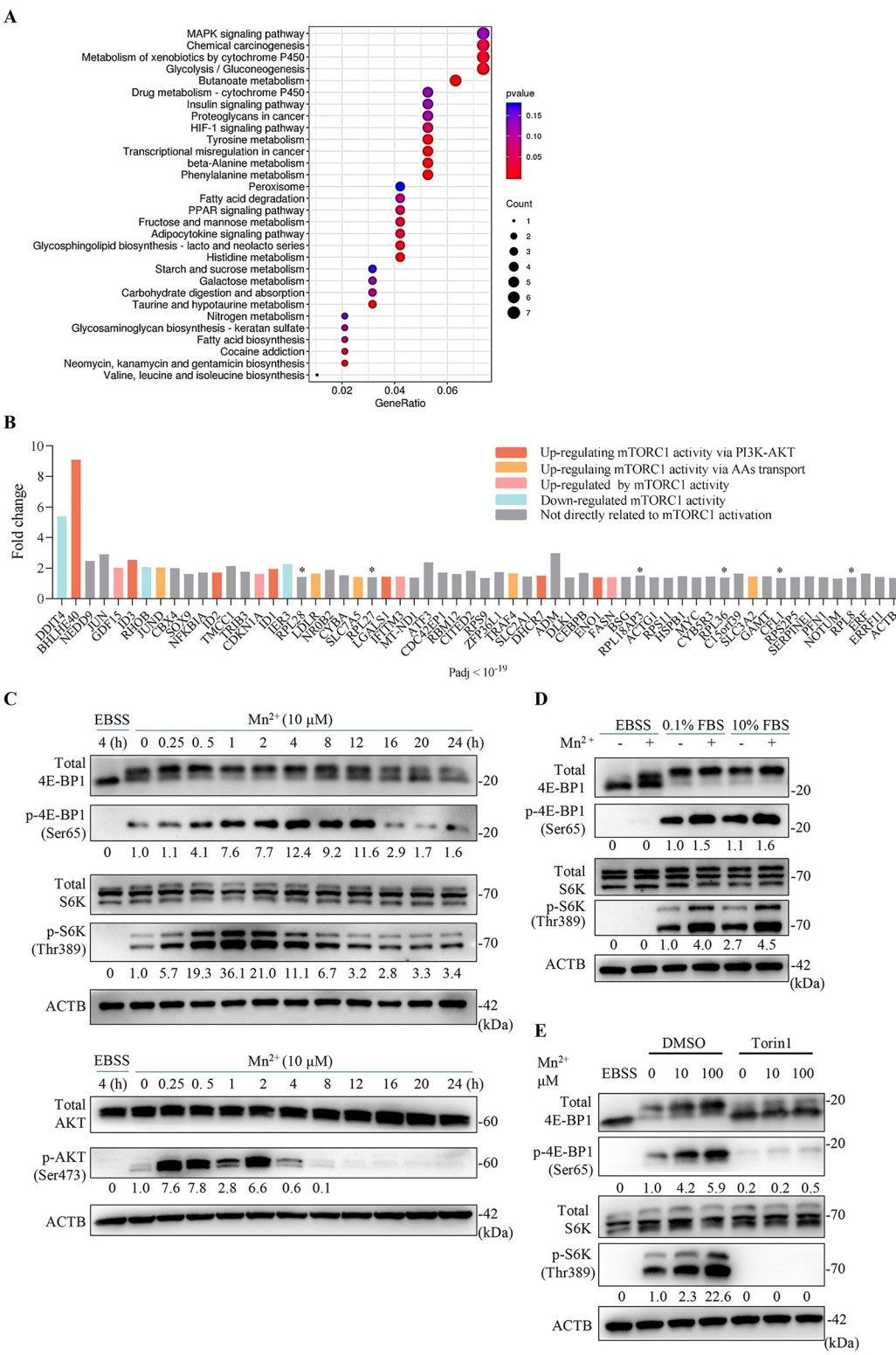

**Fig 2. Hyperactivation of mTORC1 in HepG2-NTCP cells treated with MnCl₂.** (**A**) Transcriptomic analysis performed on HepG2-NTCP cells mock-treated or treated with 10 μM MnCl₂ for 2 h. KEGG pathway enrichment analysis was used to analyze the DEGs. The size of dots indicates the proportion of genes enriched in the corresponding pathway. (**B**) Histogram showing DEGs ranked in the top 60 according to adjusted *P* values. Genes involved in regulating mTORC1 activation or regulated by mTORC1 activity are highlighted. Asterisks indicate genes encoding ribosomal components essential for protein synthesis. (**C**) Time course of mTORC1 activation and AKT phosphorylation in HepG2-NTCP cells

following exposure to $Mn^{2+}$. Cells maintained in DMEM medium containing 10% FBS were stimulated with $MnCl_2$ (10 μM) for various times, followed by immunoblotting with the indicated antibodies. The blots were analyzed by densitometry, with the intensity of the phospho-protein signal normalized to the corresponding total protein band. Cells starved in EBSS medium (amino acid deprivation) for 4 h were used as control for mTORC1 inactivation. (**D**) HepG2-NTCP cells maintained in DMEM medium containing 10% FBS or 0.1% FBS were stimulated with or without $MnCl_2$ (10 μM) for 2 h, followed by immunoblotting with the indicated antibodies. Cells starved in EBSS medium (for 4 h) were used as control. Note that in the setting of EBSS starvation, $Mn^{2+}$ had a potential effect on the forms of total 4E-BP1. Each blot is a representative of at least three independent experiments. (**E**) HepG2-NTCP cells maintained in DMEM medium containing 10% FBS were stimulated with $MnCl_2$ at the indicated concentration for 2 h with or without 1 μM Torin 1, followed by immunoblotting with the indicated antibodies. Results represent two independent experiments.

## Inactivation of mTORC1 enhanced *de novo* HBV infection in HepG2-NTCP cells

Next, we investigated whether mTORC1 regulated *de novo* HBV infection. In HepG2-NTCP cells, serum starvation (without $Mn^{2+}$ stimulation) resulted in substantial inhibition of mTORC1 activity, which coincided with a notable increase in HBV infection efficiency (**Figs 3A and S4A**). Similar observations were made in cells treated with mTORC1 specific inhibitors Torin1 and PP242 during HBV infection (**Figs 3B and S4B**), suggesting that inactivation of mTORC1 promotes HBV infection.

mTORC1-regulated ULK1/PI3K (class III) activation regulates autophagy initiation and endocytic trafficking [30, 31]. However, knockdown of the core autophagy gene ATG5 using specific small interfering RNAs (siRNAs) had no appreciable inhibitory effects on HBV infection (**S4C Fig**). We then examined whether mTORC1 regulated NTCP receptor-mediated HBV internalization and trafficking. To this end, we created a construct expressing tandem NTCP-EGFP-mCherry, which undergoes a yellow-to-red fluorescence shift when transferred to acidic endolysosomal compartments (**Fig 3C**) [32]. In transfected Huh7 cells, the NTCP-EGFP-mCherry fusion protein was located primarily along the cell boundary, which is consistent with the role of NTCP as a transmembrane transporter. Interestingly, red-only puncta were observed as perinuclear clusters approximately 20–30 min post-inoculation with HBV, suggestive of a virus-triggered endocytic process of NTCP *en route* to endo/lysosomal compartments. In particular, treatment with Torin1 significantly accelerated and increased the formation of red puncta in response to HBV stimulation (**Fig 3C**). It is noteworthy that the hyperactivation of mTORC1 induced by $Mn^{2+}$ did not lead to an opposite effect (**S4D and S4E Fig**) (see also the Discussion).

The signal-transducing adaptor molecule (STAM) have been implicated in HBV infection [33]. STAM and the hepatocyte growth factor-regulated tyrosine kinase substrate (HRS) proteins constitute the endosomal sorting complex required for transport (ESCRT)-0. We found that knockdown of ESCRT-0 subunit HRS or the ESCRT-I subunit TSG101 not only decreased HBV-stimulated lysosomal trafficking of NTCP-EGFP-mCherry (**Fig 3D**), but also largely abolished viral infection in HepG2-NTCP cells (**Fig 3E and 3F**). Of note, knockdown of HRS or TSG101 had limited effects on intracellular HBV replication in HepAD38 tet-off cells (**S4F Fig**). Interestingly, serum starvation or Torin1 treatment led to a notable increase in HRS recruitment to RAB5-positive endosomal structures, without significantly increasing the expression of these proteins (**Fig 3G**). These findings corroborate a notion that mTORC1 activity coordinates endocytic trafficking of HBV virions involving the ESCRT machinery.

## HBV infection did not depend on lysosomal fusion-driven late endosome maturation

Considering the crucial involvement of ESCRT complexes in the biogenesis of MVB/late endosomes [20], it is speculated that invading HBV virions, along with NTCP receptors, are sorted

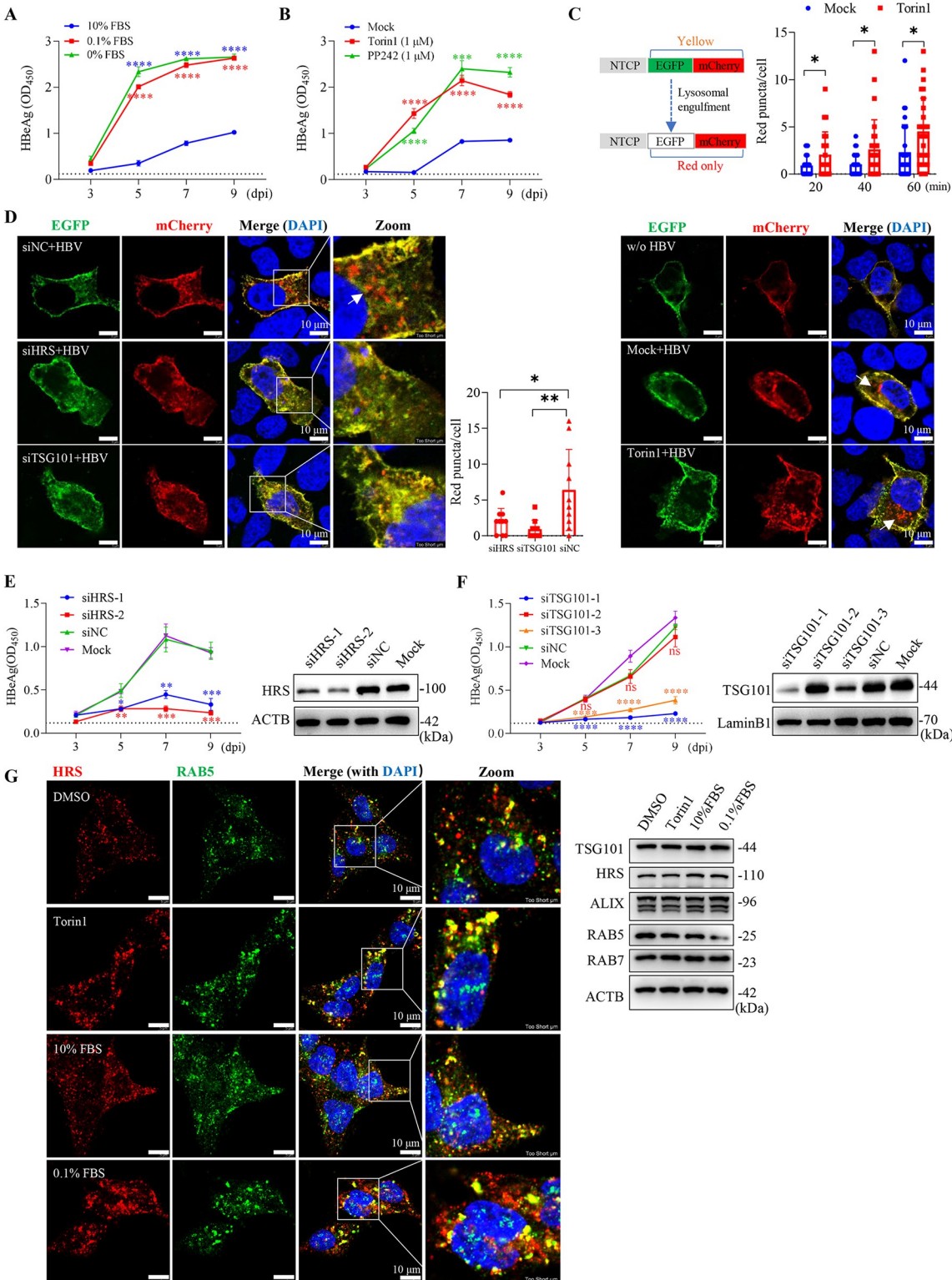

**Fig 3. Suppression of mTORC1 promotes *de novo* HBV infection in HepG2-NTCP cells.** (**A**) HepG2-NTCP cells were maintained in DMEM medium containing indicated concentrations of FBS for 10 h, followed by a further 12 h incubation with HBV inoculum (still with FBS at the indicated concentrations). Supernatant HBeAg was determined by ELISA at the indicated time points (*n* = 3). (**B**) HepG2-NTCP cells cultured in DMEM medium supplemented with 10% FBS were treated with either Torin1 (1 μM) or PP242 (1 μM) for 10 h, followed by infection with HBV in the presence of the inhibitors. The cells were then maintained in fresh medium (without

inhibitors) for subsequent HBeAg analysis (*n* = 3). Control, cells were mock-treated with DMSO. (**C**) Huh7 cells transiently transfected with a NTCP-EGFP-mCherry–expressing plasmid were treated with or without 1 μM Torin1 for 4 h, and then incubated with HBV inoculum for the indicated times. The subcellular distribution of the NTCP fusion protein was monitored by confocal microscopy. Red-only puncta are within acidic endolysosomal compartments because of the quenching of GFP fluorescence (lower, arrows). Upper left, the principle of NTCP-EGFP-mCherry to monitor the lysosomal engulfment. Upper right, the number of red dots in each cell was calculated in 8–10 fields of view over three independent experiments. (**D**) Huh7 cells transiently expressing NTCP-EGFP-mCherry were transfected with siRNA against HRS or TSG101 for 48 h. Cells were then stimulated with HBV inoculum for an additional 1 h before confocal microscopy examination. (**E and F**) HepG2-NTCP cells were transfected with siRNA against HRS (**E**) or TSG101 (**F**). After 48 h, cells were infected with HBV for an additional 12 h. Left, secreted HBeAg was determined by ELISA at the indicated time points (*n* = 3). Cells transfected with siNC (negative control siRNA) or mock transfected were used as controls. Right, the knockdown efficiency of individual siRNA was assessed by immunoblotting. (**G**) HepG2-NTCP cells were treated with Torin1 (1 μM) or serum starvation for 4 h. Left, representative images for the subcellular colocalization of endogenous RAB5 (green) and HRS (red) assessed by confocal immunofluorescence microscopy. Right, cells were analyzed by immunoblotting with the indicated antibodies. All the data shown are representative results from at least three independent experiments. Error bars indicate mean ± SD. *, $P < 0.05$; **, $P < 0.01$; ***, $P < 0.001$;****, $P < 0.0001$; ns, not significant.

into late endosomal compartments to initiate infection. RAB7 GTPase is involved in late endosome maturation and lysosomal fusion. Consistent with the previous report using the HepaRG model [10], RAB7 knockdown largely reduced HBV infection in HepG2-NTCP cells (**S5A Fig**). However, we found that viral infection was not influenced by VAMP7 knockdown (**Fig 4A**), a R-SNARE protein that governs the heterotypic fusion between late endosomes and lysosomes [34, 35], which appears to contradict the result of the RAB7 depletion experiment. The knockdown of VAMP7 resulted in a significant accumulation of LysoTracker-positive dot-like structures in stained cells, presumably representing immature MVB/late endosomes (**Fig 4B**). Nevertheless, owing to the lack of lysosomal fusion, they were unable to develop into mature LysoSensor-positive endolysosomal hybrids with further acidification [3]. Note that Lysotracker labels acidic organelles nonspecifically, whereas the LysoSensor probe is capable of track pH shifts and sensitive to the highly acidic endolysosomal vacuoles [36]. Interestingly, VAMP7 knockdown had limited effect on $Mn^{2+}$-induced hyperactivation of mTORC1 (**S5B Fig**); in HepG2-NTCP cells pre-transfected with siVAMP7, $Mn^{2+}$ still significantly inhibited *de novo* HBV infection (**S5C Fig**).

VAMP7 and syntaxin 7 (STX7) are complementary SNAREs involved in late endosome-lysosome fusion [35]. Consistent with the VAMP7 knockdown findings, STX7 knockdown impeded late endosome maturation but did not affect HBV infection (**S5D** and **S5E Fig**).

It is thought that late endosomes undergo a constant round of 'kiss and run' fusion with lysosomes [37,38]. PIKfyve is a phosphoinositide kinase whose inhibition can reduce the rate of lysosome 'run' events relative to lysosome 'kiss' events [39]. Surprisingly, the application of PIKfyve-specific inhibitors Apilimod or YM201636 markedly suppressed HBV infection in HepG2-NTCP cells (**Fig 4C**), suggesting that excessive lysosomal fusion was even detrimental to HBV infection.

Chloroquine (CQ) impairs lysosomal fusion, making it a tool compound for studying lysosomal defects [40,41]. Interestingly, HBV infection in HepG2-NTCP cells was substantially increased by CQ treatment (**Fig 4D**). Similar observations were made in cells treated with weak-base ammonium chloride ($NH_4Cl$) and the v-ATPase inhibitor bafilomycin A1 (BafA1) (**Figs 4E** and **S5F**). This result seems to contrast with the previous finding in HBV-infected differentiated HepaRG cells [10], most likely due to the different protocols used for $NH_4Cl$ administration (**S5G Fig**). Thus, impairment of endosomal/lysosomal acidification may facilitate early HBV infection. Endosome-lysosome fusion is not required for early HBV infection, suggesting that virions may escape directly from late endosome compartments to gain access to the cytosol.

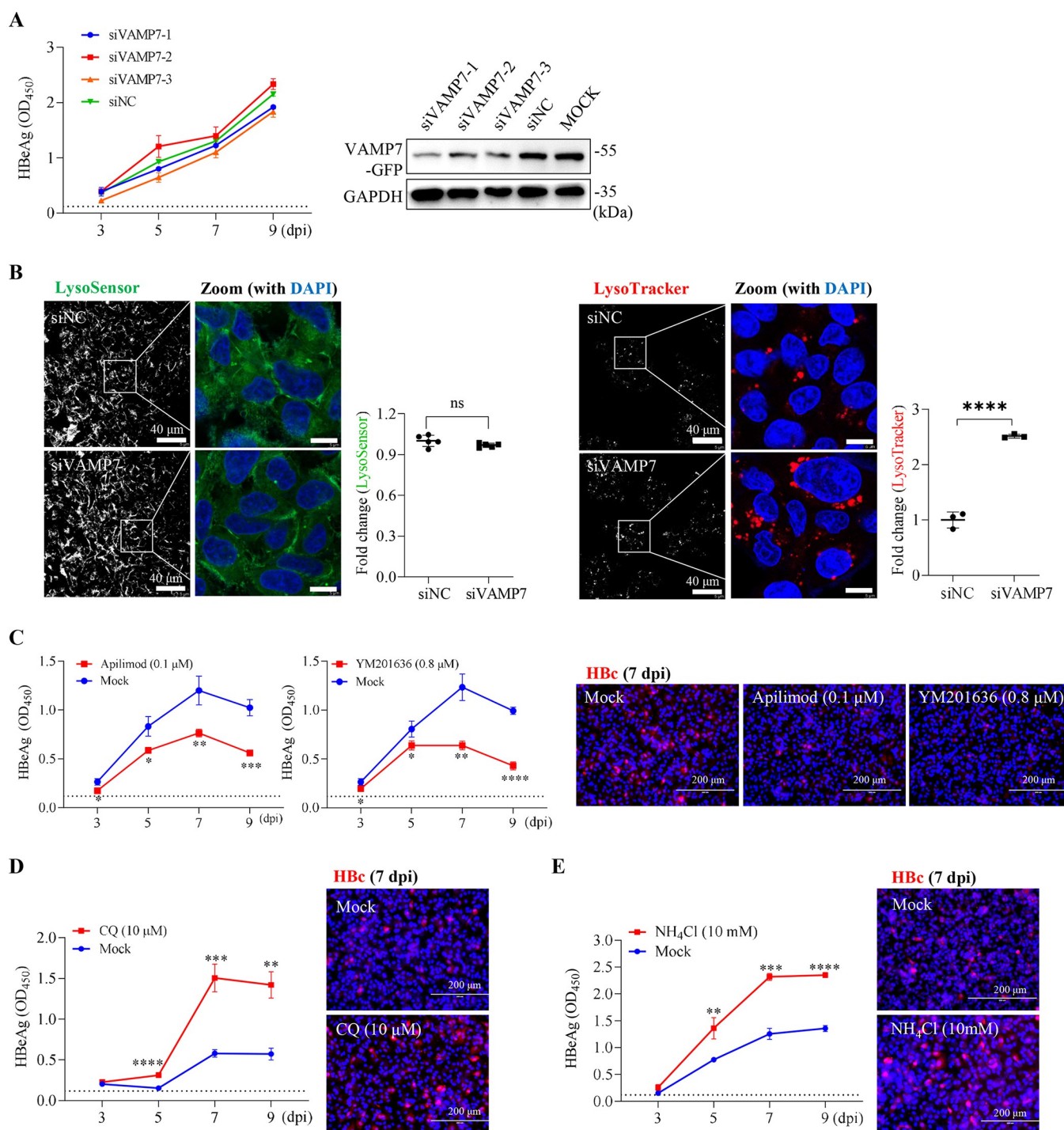

**Fig 4. Endosome-lysosome fusion is not required for early HBV infection.** (**A**) HepG2-NTCP cells were transfected with siRNAs against VAMP7 for 48 h, followed by HBV infection for an additional 12 h. Left, supernatant HBeAg was determined by ELISA at the indicated time points (*n* = 3). Right, the knockdown efficiency of individual siRNA was assessed by immunoblotting. (**B**) HepG2-NTCP cells were transfected with siVAMP7-1 for 48 h. Representative images of cells stained with LysoSensor (left) and LysoTracker (right) probes. The integrated density of five or three low-magnification (40×) fields of view was calculated using ImageJ. The results shown are representative of three independent experiments. (**C**) HepG2-NTCP cells were treated with Apilimod or YM201636 at the indicated doses for 10 h, followed by HBV infection for an additional 12 h in the presence of the inhibitors. Cells were then maintained in fresh media (without inhibitors) for subsequent analysis. The supernatant HBeAg was determined by ELISA at the indicated time points (*n* = 3). Right, representative immunofluorescence images of intracellular HBc (red) at 7 dpi. (**D and E**) HepG2-NTCP cells were treated with 10 μM CQ (**D**) or 10 mM NH$_4$Cl (**E**) for 10 h, followed by an additional 12–h incubation with HBV inoculum (in the presence of the chemicals). Supernatant HBeAg and intracellular HBc were determined as depicted in (**C**). Error bars indicate mean ± SD. *, *P* < 0.05; **, *P* < 0.01; ***, *P* < 0.001; ****, *P* < 0.0001.

## Mn$^{2+}$ was a potent inducer of lysosomal activity

Our findings raised the question of whether increased endolysosomal activity could in turn impede HBV infection. Interestingly, in addition to mTORC1 hyperactivation, we observed a notable enhancement in lysosomal activity in MnCl$_2$-treated HepG2-NTCP cells. In cells maintained in nutrient-rich medium, the lysosomal inhibitor CQ induced the accumulation of lipidated LC3 (LC3-II) puncta. Of particular interest, this effect was reversed by MnCl$_2$ treatment dose-dependently (**Figs 5A** and S6A). In the endocytic pathway, Mn$^{2+}$ enhanced epidermal growth factor (EGF)-stimulated EGF receptor (EGFR) degradation compared to sham treatment (**Fig 5B**). In line with these findings, electron microscopy imaging of MnCl$_2$-treated cells revealed a marked increase in vacuole-like degradative structures, characterized by a relatively clear content and the presence of limited lumenal materials (**Fig 5C**) [40].

We then used LysoSensor and LysoTracker probes to test endolysosomal pH and pan-acidic compartment volume, respectively. Interestingly, MnCl$_2$ treatment dose-dependently increased fluorescence intensity in cells stained with the LysoSenor probe, whereas it barely affected LysoTracker staining (**Figs 5D** and S4E). This observation was consistent with the decrease in the signal of FITC-conjugated dextran internalized into the endolysosomal lumen via micropinocytosis [42], indicating a greater level of endo/lysosomal quenching in MnCl$_2$-treated cells (**Fig 5E**). In particular, MnCl$_2$ treatment did not alter the expression levels of the lysosomal associated membrane protein 1 (LAMP1) and Cathepsin D (CTSD) (S6B Fig), suggesting that Mn$^{2+}$ did not affect *de novo* lysosome biogenesis.

Enhanced endolysosomal acidity may promote the degradation of internalized HBV virions. Therefore, we treated NTCP-EGFP–expressing Huh7 cells with MnCl$_2$ or tomatidine (see below) for 10 h. After inoculating with HBV for 1 h, NTCP-EGFP, HBsAg, and the endolysosomal marker LAMP1 exhibited partial colocalization, primarily in the perinuclear regions. As anticipated, this colocalization was significantly diminished in cells treated with MnCl$_2$. Concurrently, there was a notable reduction in intracellular puncta positive for HBsAg (S6C Fig). MnCl$_2$ treatment did not markedly reduce total NTCP protein levels during HBV infection in HepG2-NTCP cells (S6D Fig), likely due to the constitutive overexpression of NTCP, with only a small fraction undergoing HBV-triggered endocytosis and lysosomal trafficking. Altogether, these data indicate that Mn$^{2+}$ strongly stimulated lysosomal activity, which is detrimental to HBV infection.

## Inactivation of mTORC1 dynamically altered endolysosomal acidification

Inhibition of mTORC1 promotes nuclear translocation of TFEB [43], which is the primary mechanism that regulates endo-lysosomal biogenesis. Previous studies have shown that mTORC1 inhibitors promote lysosomal acidification [44]. It is not clear why mTORC1 inhibitors tend to promote HBV infection in the presence of increased lysosomal activity.

To address this issue, we stained Torin1-treated HepG2-NTCP cells with LysoTracker and LysoSensor probes and assessed the dynamics of endolysosomal acidity. After 3 h of Torin1 treatment, LysoSensor staining showed increased fluorescence intensity (**Fig 6A**), consistent with the previous report [44]. However, when treatment was prolonged to 12 h, LysoTracker staining substantially increased, although there was no corresponding change in LysoSensor fluorescence compared with untreated cells (**Fig 6B**). Similarly, Torin1 treatment initially enhanced lysosomal activity to quench FITC-dextran fluorescence; this effect however was reversed by prolonged suppression of mTORC1 (**Fig 6C**). These findings may be attributed to sustained inactivation of mTORC1, whereby excessive autophagic and endolytic activities continuously deplete lysosomes [45,46]. The marked increase in LysoTracker staining suggested that TFEB-driven *de novo* lysosome biogenesis remained active in cells after prolonged Torin1

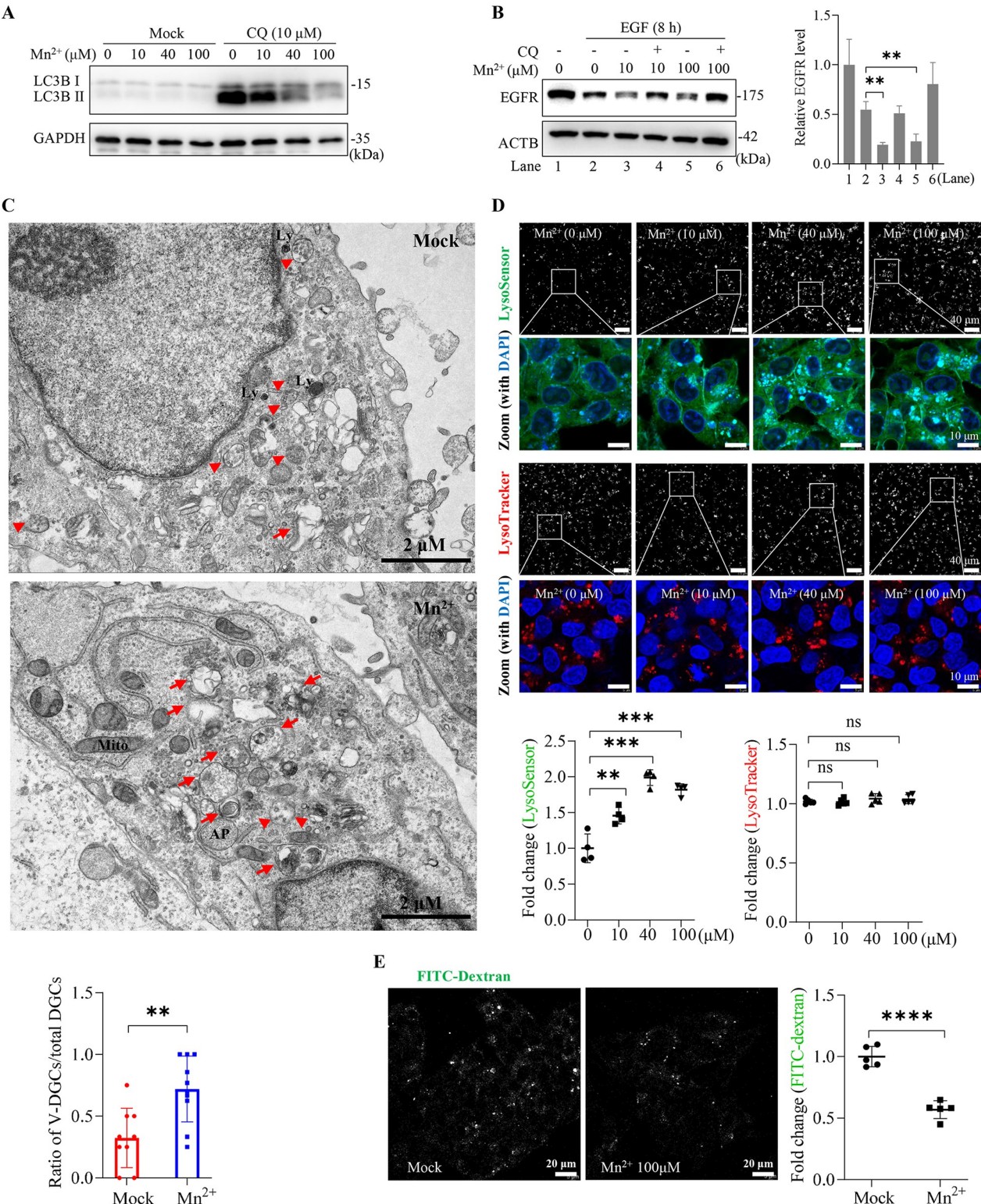

**Fig 5. Mn²⁺ promotes lysosomal function and acidification.** (**A**) HepG2-NTCP cells were treated with MnCl₂ at indicated doses for 24 h with or without CQ (10 μM). Endogenous LC3B was determined by immunoblotting. (**B**) Huh7 cells were stimulated with 10 ng/mL EGF for 8 h in the presence of MnCl₂ at the indicated doses with or without CQ (10 μM). Left, the endogenous EGFR levels were examined by immunoblotting. Right, EGFR bands were analyzed by densitometry normalized to that of ACTB. (**C**) HepG2-NTCP cells treated with or without MnCl₂ (100 μM) for 12 h were examined by transmission electron microscopy. Arrows indicate the vacuolar degradative compartments (v-DGCs). Arrowheads indicate the

nonvacuolar DGCs. Ly, lysosome; AP, autophagosome; Mito, mitochondria. Right, the ratio of v-DGCs to total DGCs per cell section was analyzed over two independent experiments with 9–10 images each. (**D**) Representative pictures of LysoSensor (upper panels) and LysoTracker (middle panels) staining of HepG2-NTCP cells treated with $MnCl_2$ at indicated doses for 12 h. The integrated density of 4–5 low-magnification (40×) fields of view was calculated using ImageJ (right). (**E**) HepG2-NTCP cells treated with or without $MnCl_2$ (100 μM) for 12 h were incubated with FITC-Dextran (5 μg/mL) for an additional 4 h, followed by confocal microscopy examination. The integrated density of five low-magnification fields of view (40×) was calculated using ImageJ. All the data shown are representative results from 2–3 independent experiments. Error bars indicate mean ± SD. **, $P < 0.01$; ***, $P < 0.001$; ****, $P < 0.0001$; ns, not significant.

treatment. However, the knockdown of the master regulator TFEB did not enhance HBV infection in HepG2-NTCP cells (**Fig 6D**). These data suggested that sustained inactivation of mTORC1 results in lysosomal dysfunction, thus facilitating early HBV infection.

## Tomatidine was analogous to $Mn^{2+}$ in its ability to hyperactivate mTORC1 and promote lysosomal function

The neurotoxicity of $Mn^{2+}$ poses a challenge for its development as an antiviral against HBV infection. Tomatidine (**Fig 7A**), a steroidal alkaloid derived from tomato plants, has been reported to exert potential therapeutic benefits against skeletal muscle atrophy by stimulating mTORC1-mediated anabolic signaling [47]. We found that in HepG2-NTCP cells, tomatidine hyperactivated mTORC1 signaling under nutrient-rich conditions, accompanied by the transient activation of AKT (**Fig 7B**). Similar to that of $Mn^{2+}$, tomatidine treatment increased EGF-triggered lysosomal degradation of EGFR (**S7A Fig**), and dose-dependently reduced the accumulation of LC3-II in CQ-treated cells (**Fig 7C**). Furthermore, lysosomal probes staining and FITC-dextran quenching assays suggested that tomatidine significantly increased the endo/lysosomal acidity of HepG2-NTCP cells (**Fig 7D** and **7E**) without apparent effects on the overall size of acidic compartments (**S7B Fig**).

V-ATPase consists of a cytoplasmic $V_1$ complex for ATP hydrolysis and a membrane-embedded $V_0$ complex for proton transfer. Regulated assembly of the two complexes enables rapid adjustment of v-ATPase activity in the endocytic pathway [48]. In HepG2-NTCP cells, tomatidine treatment markedly increased the colocalization of $V_1$ complex subunit B2 (ATP6V1B2) with the lysosomal transmembrane protein LAMP1, as examined by immunofluorescence microscopy (**Fig 7F**). This finding was further validated by immunoblot analysis, which revealed an elevated presence of ATP6V1B2 protein within the membrane fraction of tomatidine-treated cells (**Fig 7G**). Since the abundance of membrane $V_1$ subunits reflects the level of fully assembled v-ATPase complexes, these data indicate that tomatidine increases endolysosomal acidity, at least partially, by facilitating v-ATPase assembly. Like $Mn^{2+}$, tomatidine did not stimulate the *de novo* lysosomal biogenesis (**S7C Fig**). However, $Mn^{2+}$ did not stimulate the assembly of v-ATPase (**S7D Fig**), indicating that the mechanisms by which $Mn^{2+}$ and tomatidine affect endo/lysosomal function may not be identical.

## Tomatidine inhibited *de novo* HBV infection of HepG2-NTCP cells and ProliHH organoids

Tomatidine had limited cytotoxicity with an estimated CC50 of 948.2 μM in cell culture (**Fig 8A**). In HepG2-NTCP cells, tomatidine pretreatment (at doses of 3 μM or 10 μM) significantly reduced *de novo* HBV infection as evaluated by HBeAg secretion and intracellular HBc production (**Fig 8B**). Similar to $Mn^{2+}$, tomatidine pre-treatment (12 h before HBV inoculation) demonstrated the most effective inhibition compared to treatment during or post HBV infection (**Fig 8C**). Interestingly, although tomatidine did not affect HBeAg or HBsAg secretion, it caused a mild reduction in viral DNA replication, as measured by Southern blotting of

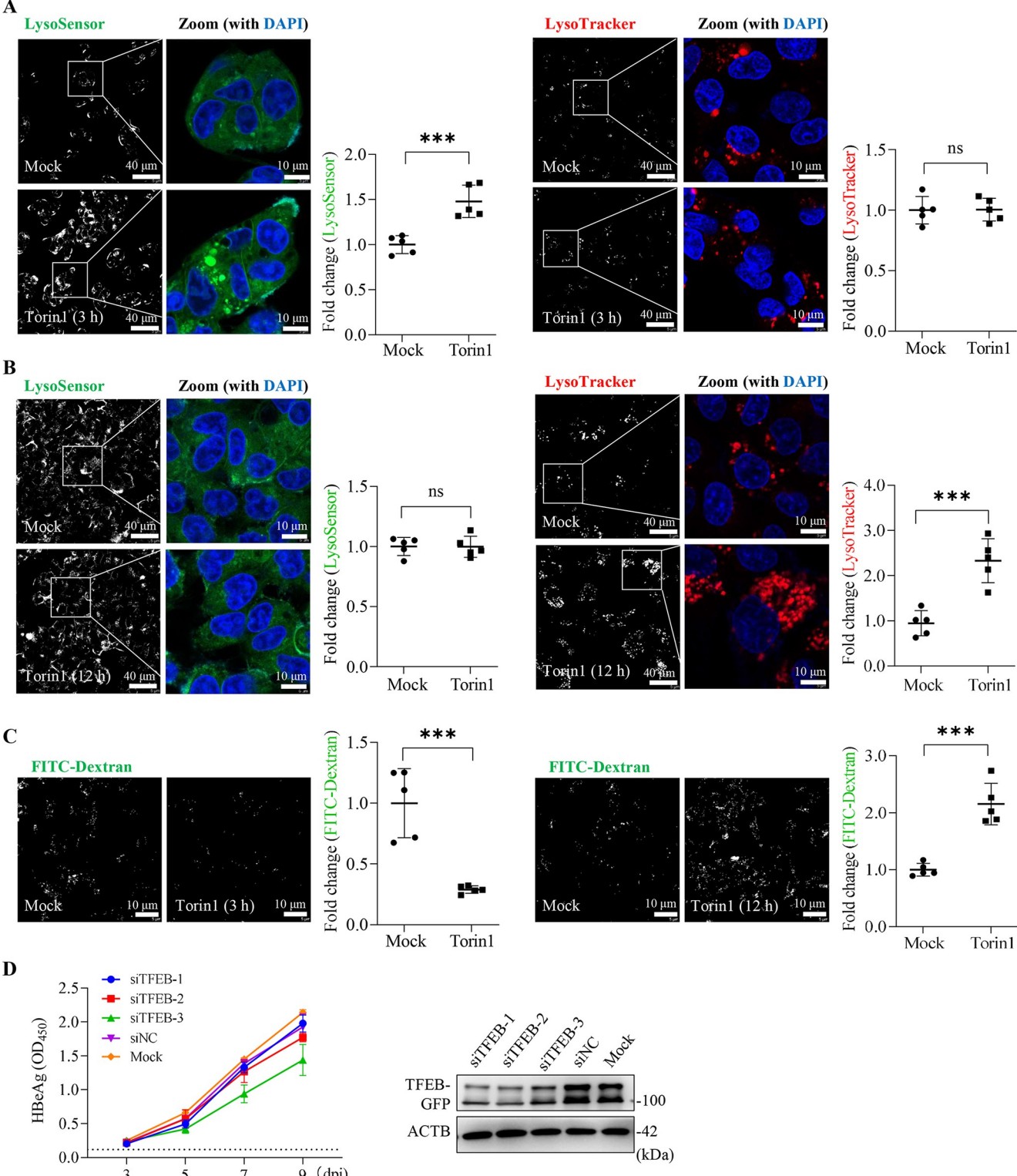

**Fig 6. Sustained inhibition of mTORC1 leads to lysosomal dysfunction.** (**A and B**) Representative images for LysoSensor (left) and LysoTracker (right) staining of HepG2-NTCP cells treated with or without Torin1 (1μM) for 3 h (**A**) or 12 h (**B**). The integrated density of five fields of view (40×) was calculated using ImageJ. (**C**) HepG2-NTCP cells treated with or without Torin1 (1μM) for 3 h (left) or 12 h (right) were incubated with FITC-Dextran (5 μg/mL) for an additional 4 h. Left, representative images of confocal microscopy. Right, the integrated density of five fields of view (40×) was calculated using ImageJ. (**D**) HepG2-NTCP cells were transfected with siRNAs against TFEB for 48 h, followed by incubation with HBV inoculum for an additional 12 h. Left, secreted

HBeAg was determined by ELISA at the indicated time points (*n* = 3). Cells transfected with siNC or mock transfected were used as controls. Right, the knockdown efficiency of individual siRNA was assessed by immunoblotting of HEK 293T cells stably expressing TFEB-GFP. All the data shown are representative results from three independent experiments. Error bars indicate mean ± SD. ***, *P* < 0.001; ns, not significant.

HepAD38 tet-off cells (**S8A** and **S8B Fig**). Thus, we generated an HBc-null HBV mutant that is capable of producing progeny virions through capsid trans-complementation, but does not support productive viral replication in infected cells. As expected, in HepG2-NTCP cells, tomatidine pre-treatment significantly reduced *de novo* infection with the replication-defective recombinant virions (**Fig 8D**).

α-tomatine (**S9A Fig**), a tomatidine derivative, activated mTORC1 (**S9B Fig**). However, α-tomatine treatment did not markedly increase endolysosomal acidification as did tomatidine treatment (**S9C Fig**). At doses close to cytotoxic levels [49], α-tomatine only weakly inhibited HBV infection of HepG2-NTCP cells (**S9D Fig**). This result suggests that the level of endo/lysosomal acidification, rather than mTORC1 hyperactivation, is the primary factor influencing the efficiency of virus infection.

There are profound differences between normal cells and transformed tumor cells. Primary human hepatocytes (PHHs) can be dedifferentiated into ProliHHs showing features of both hepatocytes and progenitor cells. After successive clonal expansions, these ProliHHs are able to regain the characteristics of mature hepatocytes in three-dimensional (3D) cultures [50,51]. The ProliHH-derived 3D organoids were susceptible to HBV infection, as indicated by the presence of intracellular HBc and secreted HBeAg over time following viral inoculation. Similar to the results observed in HepG2-NTCP cells, the treatment with lysosomal inhibitors CQ or NH$_4$Cl significantly increased the susceptibility of ProliHH organoids to HBV infection, whereas tomatidine markedly suppressed viral infection in these cells (**Fig 8E** and **8F**). Interestingly, tomatidine-treated organoids were usually larger than mock-treated organoids, possibly because of mTORC1 hyperactivation and reduced viral infection (**Fig 8F**). Of note, α-tomatine did not significantly inhibit HBV infection in ProliHH organoids (**S9E Fig**). Taken together, these data suggest that natural steroidal alkaloid tomatidine may have the potential to treat HBV infection.

## Discussion

The MVB is considered an early stage of late endosomes, and its biogenesis critically depends on ESCRT complexes. For HBV to establish an infection, it must enter MVBs/late endosomes; depletion of the ESCRT-0 and ESCRT-I subunits completely abolishes viral infection (**Fig 3E** and **3F**). HBV induces the translocation of plasma membrane-anchored NTCP into the lysosomal lumen, which also relies on the ESCRT sorting machinery (**Fig 3C** and **3D**). Myristoylated preS1 peptides retained over 50% of their binding capacity to NTCP at pH 5.5 [52]. This suggests that HBV may remain associated with NTCP in mildly acidic environments where ESCRT-mediated NTCP sorting occurs. It is unclear whether viral particles that do not enter the MVB are transported out of the cell via recycling endosomes.

In accordance with the previous report [10], our study demonstrated that RAB7 is a vital component in the process of establishing HBV infection. Given that the acidic and degradative lysosomal environment is not conducive to HBV infection, it is plausible that multifunctional RAB7 may be involved in other stages of HBV entry, rather than in the lysosomal branch. For instance, endosomal RAB7 is essential for trafficking incoming HIV capsids into the nucleus, and the underlying mechanism may be shared by other viruses that require nuclear entry to complete their life cycles [53]. The SNARE proteins VAMP7 and STX7 are dispensable for the early HBV infection (**Figs 4A** and **S5D**). These results provide further evidence that HBV

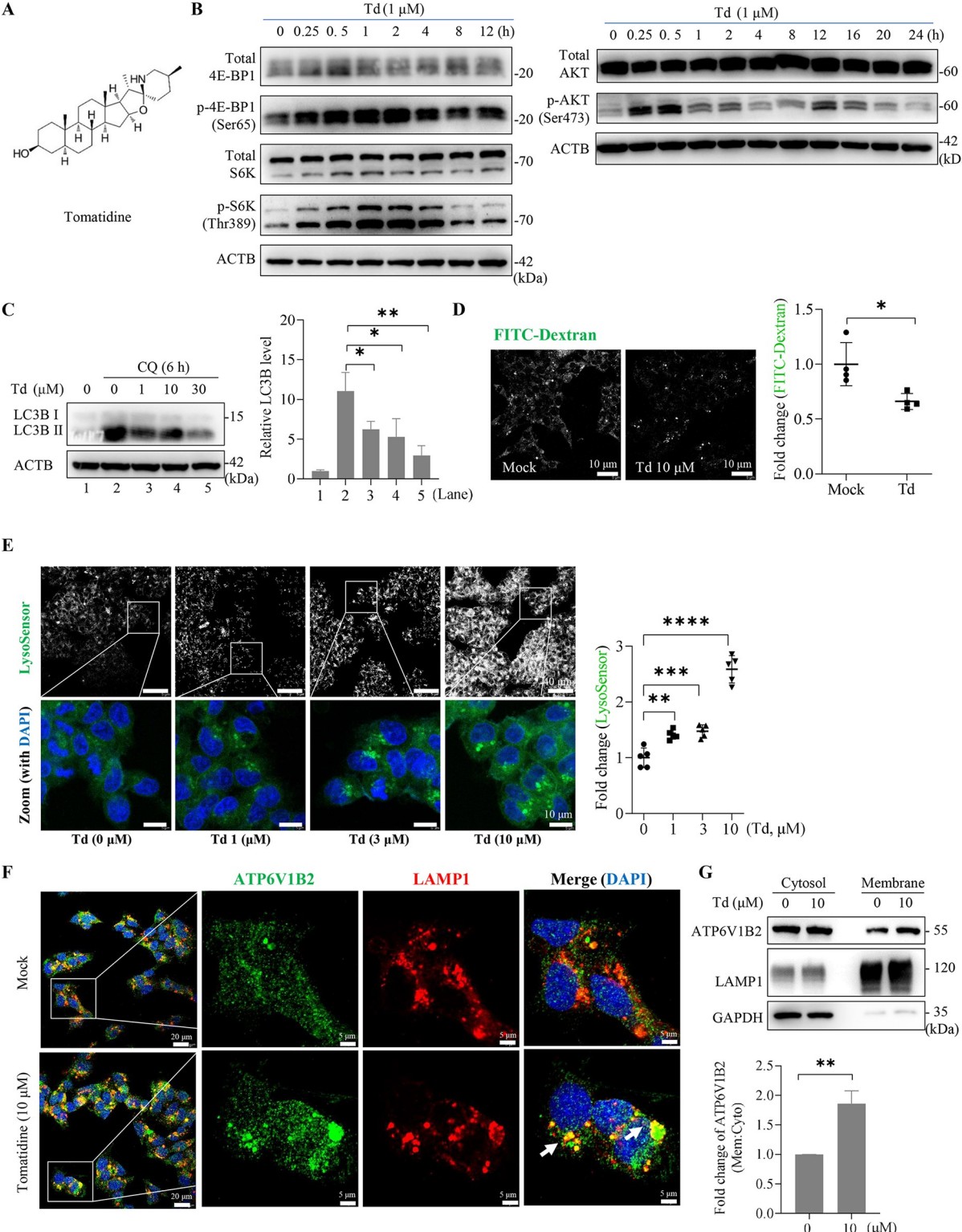

**Fig 7. Tomatidine hyperactivates mTORC1 and promotes lysosomal acidification.** (**A**) Chemical structure of tomatidine (Td). (**B**) mTORC1 activity (left) and AKT phosphorylation (right) were evaluated by immunoblotting of HepG2-NTCP cells stimulated with tomatidine (1 μM) for the indicated times. (**C**) Endogenous levels of LC3B were determined by immunoblotting of HepG2-NTCP cells treated with indicated concentrations of tomatidine for 6 h with or without CQ (10 μM). The blots were analyzed by densitometry over three independent experiments, with the intensity of the protein signal normalized to that of ACTB. (**D**) HepG2-NTCP cells treated with or without tomatidine

(10 μM) for 12 h were incubated with FITC-Dextran (5 μg/mL) for an additional 4 h, followed by confocal microscopy examination. The integrated density of four fields of view (40×) was calculated using ImageJ (right). Representative results from three independent experiments are shown. (**E**) Representative LysoSensor staining of HepG2-NTCP cells treated with indicated concentrations of tomatidine for 12 h. The integrated density of five fields of view (40×) was calculated using ImageJ (right). (**F**) HepG2-NTCP cells were treated with or without tomatidine (10 μM) for 12 h and subjected to immunofluorescence confocal microscopy. Representative images of the subcellular distribution of LAMP1 (red) and V-ATPase B2 (ATP6V1B2) (green). The DAPI-stained nuclei are in blue. (**G**) Cells were fractionated into cytosolic and crude membrane fractions, and subjected to immunoblotting with the indicated antibodies. Lysosomal LAMP1 was used as a membrane fraction loading control and GAPDH as a cytosolic loading control. All the data shown are representative results from three independent experiments. Error bars indicate the mean ± SD. *, $P < 0.05$; **, $P < 0.01$; ***, $P < 0.001$; ****, $P < 0.0001$.

achieves endosomal escape during the late endosomal stage, following ESCRT-dependent sorting of the NTCP receptor.

A recent study indicated that, as a bivalent metal cofactor, $Mn^{2+}$ is more effective than $Mg^{2+}$ in coordinating ATP in the catalytic cleft of yeast TORC1, resulting in enhanced TORC1 kinase activity [54]. It is unclear whether the described enzymatic mechanism is applicable to the mammalian counterparts of TORC1, mTORC1. Interestingly, our results indicate that $Mn^{2+}$ markedly induced AKT phosphorylation in HepG2-NTCP cells. The mTORC1 complex is subject to dual regulation: amino acids direct mTORC1 to the surface of the late endosome/lysosome, where mTORC1 becomes receptive to PI3K/AKT-mediated growth factor signaling. Thus, $Mn^{2+}$ probably induces AKT activation through MAPK/PI3K crosstalk, thereby promoting hyperactivation of mTORC1 in nutrient-rich environments. It should be noted that $Mn^{2+}$-triggered v-ATPase activation may also promote mTORC1 activity (see later discussion).

mTORC1/ULK1-regulated class III PI3K activation govern endocytic trafficking and endosome maturation [30,31,55]. The ESCRT-0 component of HRS, a downstream target of mTORC1, requires mTORC1 activity to maintain its protein expression [56]. In this study, we observed that mTORC1 inactivation promotes HBV infection in HepG2-NTCP cells. Inhibition of mTORC1 enhances HRS recruitment to early endosomes and accelerates NTCP transport to the lumen of late endosomes/endolysosomes. These findings suggest that mTORC1 regulates *de novo* HBV infection by controlling endosomal cargo transport. For invading viruses, however, accelerated endosomal trafficking may also imply faster lysosomal degradation. Interestingly, we found that prolonged mTORC1 inactivation virtually led to compromised lysosomal activity (**Fig 6A** and **6B**), most likely due to the progressive consumption of lysosomes and the concomitant impairment of mTORC1-dependent lysosomal reformation (**Fig 9**) [2,43,45].

It was previously reported that constitutively active AKT1 significantly inhibited HBV RNA transcription and, consequently, reduced HBV DNA replication in HepG2 cells [57]. However, $Mn^{2+}$-induced AKT–mTORC1 activation did not produce a comparable antiviral effect (**Figs 1D** and **S2B**), presumably due to its lower potency and shorter duration compared to the sustained activation achieved by transient transfection with the AKT mutant. It is uncertain whether $Mn^{2+}$-induced mTORC1 hyperactivation inhibits *de novo* HBV infection. mTORC1 is constitutively active in cultured cells with adequate supply of nutrients. EGF receptor (EGFR)-mediated signaling transduction is able to hyperactivate mTORC1 through PI3K/Akt–mTOR pathway [58]. Rather than inhibiting viral infection, EGF actually promotes the internalization of HBV, most likely due to the fact that EGFR may function as a co-receptor for HBV entry [33,59]. The steroidal alkaloid tomatidine and its glycoside α-tomatine are natural steroid compounds found abundantly in tissues of tomato plants. Interestingly, at subtoxic doses, α-tomatine activated AKT and therefore promoted mTORC1 hyperactivation, but did not exert significant inhibition on HBV infection (**S9 Fig**), suggesting that mTORC1 hyperactivation might not necessarily exert anti-HBV effects.

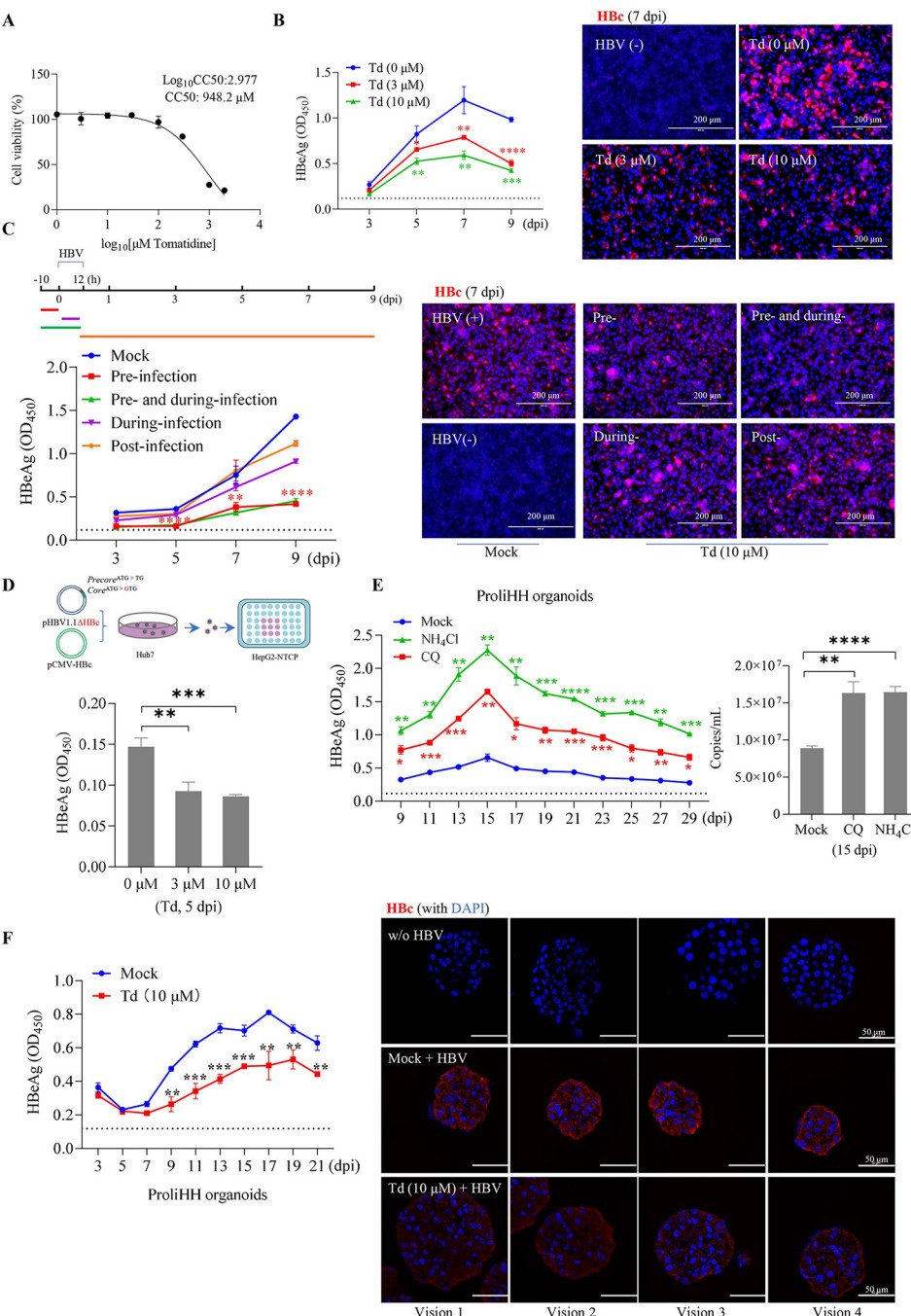

**Fig 8. Tomatidine inhibits *de novo* HBV infection in HepG2-NTCP cells and in ProliHH organoids.** (**A**) Dose response curve showing the cytotoxicity of tomatidine in HepG2-NTCP cells determined by the CCK8 assay performed in triplicate. (**B**) HepG2-NTCP cells were treated with indicated concentrations of tomatidine (Td) for 10 h, followed by HBV infection for an additional 12 h. Left, the supernatant HBeAg was determined by ELISA at the indicated time points ($n = 3$). Right, representative immunofluorescence images of intracellular HBc (red) at 7 dpi. (**C**) Time-of-addition assay of the effect of tomatidine (10 μM) on *de novo* HBV infection of HepG2-NTCP cells. Upper left, the experimental scheme. The cells were then subjected to analysis as described (**B**). Comparisons were made between pre-infection and post-infection treatment. (**D**) HepG2-NTCP cells were treated with the indicated concentrations of tomatidine for 10 h, followed by infection with a recombinant HBV rescued by HBc complementation. Upper panel, the infection scheme of the recombinant HBV complemented with HBc protein *in trans* (created using PowerPoint). Lower panel, the secreted HBeAg was determined by ELISA at dpi 5 ($n = 3$). (**E**) The ProliHH organoids were mock-treated, treated with CQ (10 μM) or NH₄Cl (5 mM) for 10 h, followed by HBV infection for an additional 12 h. The secreted HBeAg and HBV DNA in the supernatants were determined at the

indicated time points by ELISA (left) and real-time PCR (right), respectively (*n* = 3). (**F**) The ProliHH organoids were either mock-treated or treated with tomatidine (10 μM) for 10 h, followed by HBV inoculation for an additional 12 h. Left, secreted HBeAg was quantified by ELISA (*n* = 3). Right, representative images of intracellular HBc immunofluorescence staining (red) at 15 dpi. Organoids not infected with HBV served as negative controls for the labeling. Data shown are representative results from 2–3 independent experiments. Error bars indicate the mean ± SD. \*, *P* < 0.05; \*\*, *P* < 0.01; \*\*\*, *P* < 0.001; \*\*\*\*, *P* < 0.0001.

In contrast to α-tomatine, tomatidine exhibited significant inhibition of HBV infection in both HepG2-NTCP cells and non-tumor ProliHH cells. In particular, tomatidine resembles Mn²⁺ in its ability not only to induce mTORC1 hyperactivation but also to increase endolysosomal acidification, suggesting that the acidification may play a crucial role in the regulation of HBV infection. Consistently, the use of weak bases CQ or NH₄Cl, as well as the v-ATPase inhibitor BafA1, improved the efficiency of HBV infection. BafA1 treatment reportedly leads to abortive DHBV infection [11], probably because DHBV infection differs from HBV infection in the post-endocytosis step. The lack of significant cytotoxicity makes tomatidine a promising antiviral candidate.

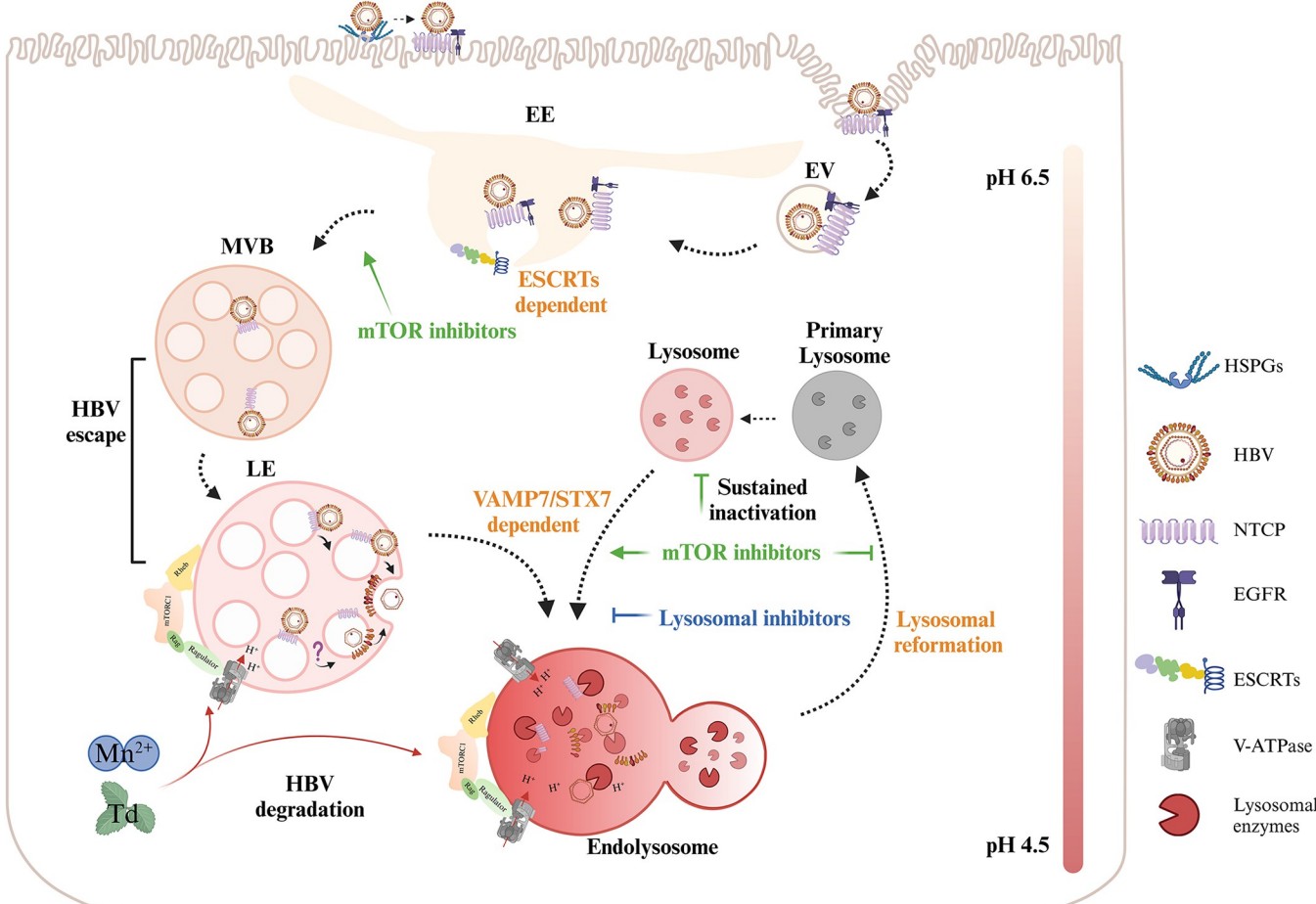

**Fig 9. Schematic diagram depicting the early stages of HBV infection and its regulation.** After HBV-triggered internalization, the NTCP receptor was sorted to late endosomal compartments by the ESCRT machinery in concert with the invading virion. The virus may gain cytosolic access directly from the late endosomes. Lysosomal hyperfunction is detrimental to early HBV infection. mTORC1 regulates *de novo* HBV infection by controlling endosomal transport. Sustained mTORC1 inactivation facilitates viral infection by depleting lysosomes. EE, early endosome. EV, endocytic vesicle; LE, late edosome. The question mark represents a molecular event that remains to be determined. The graphic was created in BioRender (https://BioRender.com/d35t466).

$MnCl_2$ is a metal ion compound, while tomatidine is a natural steroidal alkaloid. Although both compounds exhibit a common characteristic in promoting lysosomal hyperfunction, they are fundamentally distinct entities. The differences in their physicochemical properties may contribute to their divergent effects on HBV replication (**Figs 1D** and **S8A**). In addition, $Mn^{2+}$ and tomatidine may have different mechanisms for increasing lysosomal acidity. Unlike $Mn^{2+}$, tomatidine promotes the assembly of the $V_0$ and $V_1$ subunits into an intact v-ATPase (**Fig 7F** and **7G**).

The activities of v-ATPase and mTORC1 are intrinsically linked. The endo/lysosomal v-ATPase-Ragulator complex is necessary for amino acid sensing by mTORC1. V-ATPase pumps protons into the lumen of late endosomes/endolysosomes, with amino acids produced from lysosomal proteolytic digestion stimulating mTORC1 through an inside-out mechanism. Inactivation of the v-ATPase blocks the amino acid-dependent mTORC1 activation [16, 60]. Therefore, increased endo/lysosomal acidity and mTORC1 hyperactivation are both key features of lysosomal hyperfunction. However, it remains unclear why the tomatidine-derived compound α-tomatine hyperactivates mTORC1 without enhancing lysosomal acidity.

Our study primarily utilized an *in vitro* model of *de novo* HBV infection in HepG2-NTCP cells. For certain functional analyses, we also utilized ProliHH organoids, which are derived from dedifferentiated PHHs that regain mature hepatic properties after three-dimensional (3D) culture. This model offers advantages over hepatoma cell lines, as it more accurately simulates the natural HBV infection environment. However, challenges remain with the ProliHH organoid model, particularly regarding the complexity of molecular biology manipulations, such as DNA/siRNA transfection and confocal microscopy. Future investigations may still require the use of 2D-cultured PHHs to corroborate findings from our current experiments.

During our study, we also attempted to use HBV-infected chimeric mice with humanized livers to investigate the *in vivo* antiviral effects of tomatidine. Unfortunately, pharmacokinetic analyses revealed that tomatidine was rapidly cleared from the circulation after intraperitoneal or intravenous administration, complicating the attainment of effective concentrations *in vivo*. Future research may focus on exploring tomatidine derivatives or other potential lysosomal activators for their anti-HBV efficacy in humanized mouse models.

In summary, our study demonstrated that $Mn^{2+}$ and tomatidine induce lysosomal hyperfunction, thereby inhibiting *de novo* HBV infection. Following internalization, the NTCP receptor is co-transported with the HBV virion, facilitating its entry into the MVB/late endosome via an ESCRT-dependent mechanism. For effective functional infection, the virus must exit the late endosomes before significant lysosomal fusion occurs (**Fig 9**). Thus, enhancing lysosomal function and accelerating endosome-lysosome fusion represent a novel antiviral strategy for treating HBV infection.

## Materials and methods

### Cell culture

HepG2, HepG2-NTCP, HepG2-GFP-LC3B, Huh7, HeLa, HEK293T-TFEB-GFP cells were maintained in Dulbecco's modified Eagle medium (DMEM, VivaCell) supplemented with 10% fetal bovine serum (FBS), 100 U/mL penicillin, and 100 μg/mL streptomycin (VivaCell) at 37˚C containing 5% $CO_2$ incubators. HepAD38 cells were maintained in DMEM/F-12 medium supplemented with 10% FBS, 100 U/mL penicillin,100 μg/mL streptomycin, 1 μg/mL doxycycline, and 400 μg/mL G418. Removal of doxycycline from the culture medium induced the transcription of HBV pgRNA and DNA replication. The proliferating human hepatocyte (ProliHH) organoids (kindly provided by HEXAELL Biotech) were maintained in human hepatocyte induction medium supplemented with 10% FBS, 100 U/mL penicillin, and 100 μg/mL streptomycin at 37˚C containing 5% $CO_2$ incubators[50].

## Antibodies

Antibodies against Phospho-p70 S6 kinase (p-S6K Thr389,9234), phospho-4E-BP1(p-4E-BP1 Ser65,13443),4E-BP1 (9644), phospho-AKT (p-AKT, Ser473, 4060), AKT (4691), LAMP1 (19091), RAB5 (3547), RAB7 (9367), STING (13647), and EGFR (4267) were purchased from Cell Signaling Technology. Antibodies against S6K(14485-1-AP), ACTB (81115-1-RR), GAPDH (60004-1-Ig), GFP (50430-2-AP), HRS (67818-1-Ig), TSG101 (28283-1-AP), STX7 (12322-1-AP), ATG5 (66744-1-Ig), and Histone H3 (17168-1-AP) were obtained from Proteintech Group. Antibodies against LC3B (ab192890) and CTSD (ab75852) were from Abcam. Antibodies against ATP6V1B2 were purchased from Santa Cruz Biotechnology (sc-55544 for WB and sc-271832 for IF). Antibodies against NTCP were obtained from Sigma-Aldrich (HPA042727) or Abcam (ab131084). For capsids detection, a mouse monoclonal anti-HBc C1 (ab8637) was from Abcam. For immunofluorescence staining, a rabbit polyclonal anti-HBc (0282) was purchased from Long Island Antibody; a mouse monoclonal anti-HBsAg (m66) was kindly provided by Prof. Jieliang Chen of Fudan University. For western blotting, a mouse monoclonal anti-HBc 2A7 was kindly provided by Prof. Quan Yuan of Xiamen University; a rabbit polyclonal anti-HBsAg (NB100-62652) was obtained from Novus Biologicals.

## Chemicals

$MnCl_2 \cdot 4H_2O$ (M5005), Dimethyl Sulfoxide (DMSO, D2650), polyethylene glycol 8000 (PEG8000, 89510), $NH_4Cl$ (A9434) were purchased from Sigma-Aldrich. 2'3'-Cgamp (tlrl-nacga23-02) was from InvivoGen. Tomatidine (HY-N2149), chloroquine (CQ, HY-17589A), Torin1 (HY-13003), PP242 (HY-10474), Apilimod (HY-14644), YM201636 (HY-13228), entecavir (HY-13623), FITC-Dextran (MV10000, HY-128868), LysoTracker (HY-D1300) were obtained from MedChemExpress. Doxycycline hyclate (S4163) was from Selleck Chemicals. G418 (A100859) was from Sangon Biotech. The LysoSensor (MB6043) was from Meilunbio.

## Plasmids

pHBV1.3, pCMV-HBc and pCMV-HBV1.1 (the ATG start codon of *precore* gene was mutated to TG) were used in our laboratory. pCMV-HBV1.1Δ*HBc* was derived from pCMV-HBV1.1, which is defective in the expression of the HBV core, by replacing the ATG start codon of the *core* gene with GTG. pCMV-HBV1.1Δ*HBc* and pCMV-HBc were cotransfected into Huh7 cells to prepare recombinant HBV virions.

## HBV infection

HBV inoculum was obtained from HepAD38 cell culture supernatants and concentrated 100 times using precipitation of PEG8000. The HBV copy numbers were determined using quantitative PCR. For HBV infection, HepG2-NTCP cells (80% to 90% confluence) or ProliHH organoids were infected with 500 multiplicity of genome equivalents (MGE) of HBV in culture medium containing 4% PEG8000 and 2.5% DMSO for 12 h. Cells were washed five times with PBS and cultured in DMEM complete culture medium supplemented with 2.5% DMSO (without additional chemicals unless otherwise stated).

## LysoSensor and LysoTracker labelling

After the indicated treatment, cells were stained with 1 μM of LysoSensor Green for 30 min or 50 nM of LysoTracker Red for 20 min followed by Hoechst33342 incubation for another 10 min. Cells were rinsed with PBS for three times and subjected to confocal microscopy.

## Cell transfection

For DNA transfection, Neofect DNA transfection reagent (Neofect biotech Co., Ltd, TF201201) was used according to the manufacturer's instructions. Briefly, plasmids DNA diluted in Opti-MEM were mixed with Neofect (1:1, w/v) and incubated at room temperature for 20 min. The mixture was added to the medium for 8–12 hours and then cells were replaced with fresh DMEM medium.

For siRNA transfection, Lipofectamine RNAiMAX reagent (Invitrogen, 13778150) was used according to the manufacturer's instructions. Briefly, cells were seeded at approximately 80% confluence at the time of transfection. The RNAiMAX reagent and siRNA were diluted in Opti-MEM as recommended by the manufacturer. The mixture was incubated at room temperature for 10 min and added to the cell suspension. The medium was were replaced with fresh complete DMEM after 12 h.

## Immunofluorescence

After washing cells with PBS, these were fixed in 4% PFA for 10 min at room temperature. Fixed cells were permeabilized with 0.2% Triton X-100 in PBS containing 20% goat serum for 30 min at room temperature. After rinsing with PBS, cells were incubated with the indicated primary antibodies overnight at 4˚C. Cells were rinsed with PBS and incubated with Alexa Fluor-tagged secondary antibodies for 1 h. The cells were then rinsed with PBS and stained with DAPI (Beyotime Biotechnology, C1002) to stain the nuclei.

For HBV-infected ProliHHs, organoids were fixed in 4% PFA in situ for at least 4 h at room temperature. Cell pellets were collected and reserved in 4% PFA at 4˚C. Sample processing and image scanning were performed using AIMINGMED Tech. Co., Ltd, (Hangzhou, China).

## FITC-Dextran (MW10000) analysis

After the required treatments, cells were incubated with FITC-Dextran (5μg/mL) for 4 h. Washing cells with PBS for three times, these were fixed in 4% PFA for 10 min at room temperature. Cells were stained with DAPI for 5 min after washing with PBS and then subjected to confocal microscopy.

## Immunoblotting

The cells were washed with PBS and lysed in 2×SDS loading buffer. The boiled cell lysate was electrophoresed using sodium dodecyl sulfate-polyacrylamide gel electrophoresis and transferred to PVDF membrane (Merck Millipore, ISEQ00010). The membrane was blocked with 5% skim milk for 1 h at room temperature and then incubated with primary antibodies overnight at 4˚C followed by incubation with secondary antibodies (Abcam, ab6789 and ab6721) for 1 h at room temperature. Protein bands were detected using the automatic chemiluminescence image analysis system Tanon-4600 (Shanghai, China)

## Southern blotting

Capsid-associated HBV DNA was extracted as previously described, with minor modifications [61]. Briefly, cells were lysed in 200 μL lysis buffer (1% NP 40; 100 mM NaCl; 1mM EDTA pH 8.0; 10mM Hepes pH 7.5), then incubated with Mung Bean Nuclease (New England Biolabs, M0250S), DNase I (Sangon Biotech, EN0521), RNaseA (Thermo Fisher Scientific, EN0531) to remove the genomic DNA. Viral DNA was ethanol precipitated after protease K digestion in the presence of 0.5% SDS at 56˚C overnight. The purified DNA was dissolved in 20 μL double-

distilled water and separated by 1% agarose gel. Following an overnight transfer to a nylon membrane, the blot was hybridized to a digoxigenin-labeled HBV-specific probe.

### Northern blotting

Total RNA was extracted using TRIzol Reagent (Invitrogen, 15596018CN) according to the manufacturer's instructions. Equal purified RNA dissolved in DEPC water was bathed at 65˚C for 10 min, rapid ice bathed for 1 min, and then separated by 1.5% agarose gel containing 2.2M formaldehyde and MOPS. Following overnight transfer to nylon membrane, the blot was hybridized to a digoxigenin-labeled HBV-specific probe.

### Particle gel assay

Intracellular HBV capsids, along with virions and naked capsids in the supernatant, were detected via particle gels and immunoblotting, as previously described [62].

### Electron microscopy

After treatment, cells were gently scraped and centrifuged at 800 rpm for 3 min. The cell pellet was fixed with a fixative containing osmium tetroxide and potassium ferrocyanide for 30 min. Sample processing and image scanning were performed by ServiceBio (Wuhan, China).

### Cell fractionation

For nuclear separation from cytosolic fractions, cells were rinsed with ice-cold PBS and scraped into to 100 μL ($2\times10^6$ of cells) CSK buffer (10 mM PIPES pH 6.8, 100 mM NaCl, 300 mM sucrose, 3 mM $MgCl_2$, 1 mM EDTA, 0.5% Triton X-100) and lysed on ice for 15 min. Cell suspensions were centrifuged at 1,699 *g* for 5 min at 4˚C and supernatants were collected as cytosolic fractions. The pellets were resuspended with 50 μL CSK buffer and the supernatants were discarded after centrifugation at 1,699 *g* for 5 min at 4˚C. Nuclear pellets were solubilized in 1×SDS loading buffer.

A commercial kit was used to separate membranes from cytosol fractions (Beyotime Bio-technology, P0033). Briefly, $5\times10^7$ cells were rinsed with ice-cold PBS, scraped into 1 mL buf-ferA (containing 1 mM PMSF), and lysed on ice for 15 min. The cell suspensions were ground at least 50 times, and the crude lysates then were cleared of unbroken cells and nuclei by centrifugation at 700 *g* for 10 min at 4˚C. The cleared supernatants then were centrifuged at 14,000 *g* for 30 min at 4˚C to sediment the membrane fractions and collect supernatants as cytosolic fractions. The membrane pellets were solubilized in 200 μL of bufferB (containing 1 mM PMSF) and ice-bathed for 15 min. Nuclear suspensions were centrifuged at 14,000 *g* for 5 min and supernatants were collected as nuclear fractions.

### ELISA

The supernatant HBeAg, HBsAg (Kehua Bio-Engineering co., Ltd.) and IFN-β (Proteintech Group, KE00187) were determined using commercial enzyme-linked immunosorbent assay kits according to the manufacturer's instructions.

### Cell viability assay

Cell viability was detected using a commercial Enhanced Cell Counting Kit-8 (Beyotime Bio-technology, C0041) according to the manufacturer's instructions.

## Real-Time quantitative PCR

Supernatant HBV DNA was extracted using a commercial kit according to the manufacturer's instructions (Vazyme, RC311). Absolute real-time PCR was performed using the PowerTrack SYBR Green Master Mix (Thermo Fisher Scientific, A46012). To determine the relative level of intracellular mRNA, total RNA was extracted using TRIzol Reagent according to the manufacturer's instructions. First-strand cDNA synthesis by reverse transcription was performed using the FastKing-RT SuperMix kit (TIANGEN, KR118). The relative level of RNA expression level was quantified using PowerTrack SYBR Green Master Mix, with relative amounts calculated using the $2^{-\Delta\Delta Ct}$ method (normalized to GAPDH).

## Statistical analysis

Data were presented as mean ± standard deviation (SD). Statistical analysis was performed with Student's *t* test or two-way analysis of variance (ANOVA) with Tukey's multiple-comparison test when the experimental design was more than two groups. GraphPad Prism 8 (La Jolla, CA, USA) was used to analyze the data.

For sequences for siRNA, PCR and RT-qPCR primer, please refer to the supplementary information.

## Supporting information

**S1 Fig. $Mn^{2+}$ does not stimulate the STING-mediated type I interferon response in HepG2 cells.** (**A**) The STING protein was determined in HeLa, Huh7 or HepG2 cells by immunoblotting. (**B**) HepG2 cells were treated with the indicated concentrations of $MnCl_2$ for 24 h, followed by incubation in fresh medium for an additional 24 h. The supernatant IFN-β was analyzed by ELISA ($n = 3$). Control, cells treated with cGAMP (10 μg/mL). (**C**) HepG2 cells were treated with indicated concentrations of $MnCl_2$ or cGAMP for 8 h, followed by quantitative reverse transcription PCR using *Mx1-*, *ISG15-*, *OAS1*-specific primers ($n = 3$). Values show the mean ± SD. *, $P < 0.05$; ns, not significant.
(TIF)

**S2 Fig. $MnCl_2$ treatment does not inhibit HBV replication and viral egress.** (**A**) HepG2-NTCP cells were treated with or without 10 μM $MnCl_2$ for 10 h, followed by HBV inoculation as depicted in **Fig 1A**. Secreted HBeAg was determined by ELISA at the indicated time points ($n = 3$). (**B**) HBV replication was induced in HepAD38 cells by removing doxycycline (tet-off) for 72 h. Cells were then treated with $MnCl_2$ at indicated concentrations for an additional 72 h, followed by Southern blotting for intracellular viral DNA species, northern blotting for viral RNA transcripts, immunoblotting for viral proteins, and the particle gel assay for viral capsids, respectively. (**C**) Quantitative PCR analysis of intracellular HBV DNA (left) or RNA (right) in HepAD38 cells treated with $MnCl_2$ as depicted in (B) ($n = 3$). (**D**) ELSIA of HBeAg or HBsAg in the supernatant of HepAD38 cells as depicted in (B) ($n = 3$). (**E**) Huh7 cells transfected with pHBV1.3 were treated with 40 μM $MnCl_2$ (left panels) or 10 μM tomatidine (Td, right panels) in fresh medium for 72 h. Viral particles in the supernatant were concentrated by PEG8000 precipitation and resolved by agarose gel electrophoresis. Enveloped virions and naked capsids were detected by immunoblotting with the indicated antibodies. Virion/capsid-associated DNA was determined by quantitative real-time PCR ($n = 3$). Error bars indicate the mean ± SD. *, $P < 0.05$; **, $P < 0.01$; ***, $P < 0.001$; ****, $P < 0.0001$; ns, not significant.
(TIF)

**S3 Fig. MnCl₂ treatment does not affect NTCP expression and subcellular distribution.**
(**A**) HepG2-NTCP cells were treated with the indicated concentrations of MnCl₂ for 4, 8, or 24 h, followed by immunoblotting with anti-NTCP. HepG2 cells served as negative control. (**B**) HepG2-NTCP cells were treated with indicated concentrations of MnCl₂ for 8 h. Biotin-labeled cell surface proteins were precipitated on streptavidin-agarose beads and subjected to immunoblotting. (**C**) Immunofluorescence confocal microscopy of the subcellular distribution of NTCP in HepG2-NTCP cells treated with MnCl₂ at the indicated concentrations for 12 h. Scale bar = 5 μm. Blue, DAPI; Green, NTCP.
(TIF)

**S4 Fig. Serum starvation and mTORC1 inhibitors facilitate *de novo* HBV infection of HepG2-NTCP cells.** (**A**) HepG2-NTCP cells were cultured in DMEM medium containing FBS at the indicated concentrations for 10 h, followed by a further 12-h incubation with HBV inoculum. Left, representative immunofluorescence images of intracellular HBc (red) at 7 dpi. Right, mTORC1 activity was assessed by immunoblotting of phosphorylated S6K and 4E-BP1. (**B**) HepG2-NTCP cells cultured in DMEM medium with 10% FBS were treated with Torin1 (1 μM) or PP242 (1 μM) for 10 h, followed by another 12-h incubation with HBV inoculum (in the presence of the inhibitors). Cells were examined as described in (**A**). Control, cells mock-treated with DMSO. (**C**) Knockdown of ATG5 does not suppress *de novo* HBV infection. HepG2-NTCP cells were transfected with ATG5-specific siRNAs for 48 h, followed by incubation with HBV inoculum for 12 h. Top, secreted HBeAg was determined by ELISA at the indicated time points ($n$ = 3). The dotted line represents the cutoff value. Bottom, the knockdown efficiency of each siRNA was assessed by immunoblotting. siNC, negative control siRNA. (**D**) Huh7 cells were transiently transfected with a plasmid expressing NTCP-EGFP-mCherry and treated with or without 40 μM MnCl₂ for 2 h (corresponding to peak mTORC1 activation (Fig 2C), although lysosomal acidification had not yet increased (see below)). The cells were then inoculated with HBV for 1 h and analyzed by confocal microscopy (left). Red puncta in each cell were counted across 30 fields of view in two independent experiments (right). (**E**) Representative images of LysoSensor-stained Huh7 cells treated with 40 μM MnCl₂ for the indicated durations are shown (left). Integrated density from five low-magnification (40×) fields of view was quantified using ImageJ (right). (**F**) HepAD38 tet-off cells were transfected with siRNA against HRS or TSG101 for 3 days. Intracellular viral DNA was detected by Southern blotting. RC, relaxed circular DNA; DSL, double-stranded linear DNA; SS, single-stranded linear DNA.
(TIF)

**S5 Fig. Lysosomal fusion is dispensable for establishing HBV infection.** (**A**) HepG2-NTCP cells were transfected with RAB7-specific siRNAs for 48 h, followed by incubation with HBV inoculum for 12 h. Left, secreted HBeAg was determined by ELISA at the indicated time points ($n$ = 3). Right, the knockdown efficiency of each siRNA was assessed by immunoblotting. siNC, negative control siRNA. (**B**) 48 h after siVAMP7-2 transfection, HepG2-NTCP cells were treated with MnCl₂ (10 μM) for the indicated durations, followed by immunoblotting with the specified antibodies. The blots were analyzed by densitometry, and the intensity of the phospho-protein signals was normalized to the corresponding total protein bands. Cells starved in EBSS medium for 4 h were used as a control for mTORC1 inactivation. (**C**) HepG2-NTCP cells transfected with siVAMP7-2 (left) or siNC (right) were treated with 40 μM MnCl₂ for 10 h, followed by an additional 12 h incubation with HBV inoculum. Secreted HBeAg was determined at the indicated time points ($n$ = 3). (**D**) HepG2-NTCP cells were transfected with STX7-specific siRNAs for 48 h and analyzed as described in (**A**). (**E**) HepG2-NTCP cells were transfected with siSTX7-2 for 48 h. Representative images of cells

stained with LysoSensor (top) and LysoTracker (bottom) probes are shown. The integrated density of 9–10 low-magnification (40×) fields of view was calculated using ImageJ. (**F**) HepG2-NTCP cells were treated 10 nM BafA1 for 10 h, followed by HBV infection for an additional 12 h. Left, the supernatant HBeAg was measured by ELISA at the indicated time points (*n* = 3). Right, representative immunofluorescence images of intracellular HBc (red) at 7 dpi. (**G**) HepG2-NTCP cells were inoculated with HBV for 12 h and then cultured in fresh medium for an additional 24 h. Cells were subsequently treated with $NH_4Cl$ at the indicated concentrations for another 24 h. Top, experimental scheme. Bottom left, secreted HBeAg was determined at the indicated time points (*n* = 3). Bottom right, representative images of immunofluorescence labeling of intracellular HBc at 7 dpi. Error bars indicate the mean ± SD. *, $P < 0.05$; **, $P < 0.01$; ***, $P < 0.001$; ****, $P < 0.0001$; ns, not significant. (TIF)

**S6 Fig. $Mn^{2+}$ counteracts the lysosomal inhibitory effect of CQ.** (**A**) HepG2 cells that stably express GFP-LC3B were treated with indicated concentrations of $MnCl_2$ for 24 h with or without 10 µM of CQ. Left, GFP-LC3B determined by immunoblotting with anti-GFP. Right, representative images of GFP-LC3B by immunofluorescence confocal microscopy. (**B**) HepG2-NTCP cells were treated with indicated concentrations of $MnCl_2$ for 12 h. Left, endogenous LAMP1 and cleaved CTSD determined by immunoblotting. Right, subcellular distribution of LAMP1 assessed by immunofluorescence confocal microscopy. (**C**) NTCP-EGFP–expressing Huh7 cells were treated with $MnCl_2$ (40 µM) or tomatidine (Td, 10 µM) for 10 h. After inoculation with HBV for 1 h, cells were stained with anti-HBsAg and anti-LAMP1, and analyzed by immunofluorescence confocal microscopy. Left, representative images from two independent experiments. White puncta (arrows) indicate the colocalization of HBsAg (pink), LAMP1 (red), and NTCP-EGFP (green). Right, the intracellular pink puncta, as well as the white puncta resulting from the merging of the three colors, were quantified per cell from a total of 25 cells in each group. (**D**) HepG2-NTCP cells were pre-treated with $MnCl_2$ at the indicated concentrations for 10 h, followed by the stimulation with HBV inoculum for either 1 h or 12 h (in the continued presence of $MnCl_2$). NTCP expression was then assessed by immunoblotting. Error bars indicate the mean ± SD. ***, $P < 0.001$; ****, $P < 0.0001$. (TIF)

**S7 Fig. Tomatidine promotes lysosomal function** (**A**) Huh7 cells were stimulated with EGF (10 ng/mL) for 8 h in the presence of tomatidine (Td) at the indicated concentrations, with or without CQ (10 µM). The level of endogenous EGFR was determined by immunoblotting. The blots were analyzed by densitometry, with the intensity of the EGFR band normalized to the corresponding ACTB band. Representative data are shown from three independent experiments. (**B**) Representative LysoTracker staining of HepG2-NTCP cells treated with or without tomatidine (10 µM) for 12 h(left). The integrated density of five fields of view (40×) was calculated using ImageJ (right). (**C**) HepG2-NTCP cells were treated with or without tomatidine (10 µM) for 12 h. Left, endogenous LAMP1 and cleaved CTSD determined by immunoblotting. Right, subcellular distribution of LAMP1 assessed by immunofluorescence confocal microscopy. (**D**) HepG2-NTCP cells were fractionated into crude cytosolic and membrane fractions after 12 h of treatment with or without $MnCl_2$ (100 µM), and then subjected to immunoblotting with the indicated antibodies. Lysosomal LAMP1 was used as a membrane fraction loading control and GAPDH as a cytosolic loading control. Error bars indicate the mean ± SD. ns, not significant. (TIF)

**S8 Fig. Tomatidine modestly inhibits HBV replication in HepAD38 cells without affecting viral antigen secretion.** (**A**) HBV replication was induced in HepAD38 cells by the removal of

doxycycline (tet-off) for 72 h. Cells were then treated with tomatidine (Td) at indicated concentrations for an additional 72 h, followed by Southern blotting for intracellular viral DNA species. RC, relaxed circular DNA; DSL, double-stranded linear DNA; SS, single-stranded linear DNA. Cells treated with 2 μM entecavir (ETV) were as a control. (**B**) Huh7 cells transiently transfected with pHBV1.3 were treated with different concentrations of tomatidine for 3 days. HBeAg and HBsAg in the supernatants were determined by ELISA ($n$ = 3). Error bars indicate mean ± SD. ns, not significant.
(TIF)

**S9 Fig. α-Tomatine does not have an inhibitory effect on *de novo* HBV infection.** (**A**) Chemical structure of α-Tomatine (α-Tt). (**B**) HepG2-NTCP cells maintained in DMEM medium containing 10% FBS were stimulated with α-tomatine (1 μM) for various times, followed by immunoblotting with the indicated antibodies. Cells starved in EBSS medium (without amino acids) for 4 h were used as a control for mTORC1 inactivation. (**C**) Representative LysoSensor staining of HepG2-NTCP cells treated with or without α-tomatine (1 μM) for 12 h. (**D**) HepG2-NTCP cells were treated with or without α-tomatine (0.5 μM) for 10 h, followed by HBV infection for an additional 12 h. The supernatant HBeAg was determined by ELISA at the indicated time points ($n$ = 3). Note that treatment with 1 μM α-tomatine slightly reduced the viability of HepG2-NTCP cells. (**E**) ProliHH organoids were mock-treated or treated with α-tomatine for 10 h, followed by HBV infection for an additional 12 h. ProliHHs were more resistant to α-tomatine-induced cytotoxicity than tumor cells at 1 μM and 3 μM. The secreted HBeAg and HBV DNA in the supernatants were determined by ELISA (left) and real-time PCR (right), respectively, at the indicated time points ($n$ = 3). Error bars indicate mean ± SD. ns, not significant.
(TIF)

**S1 Table. siRNA sequences.**
(DOCX)

**S2 Table. Primer sequences for PCR and RT-qPCR.**
(DOCX)

**S1 Dataset. Figs 1–9 dataset.**
(ZIP)

**S2 Dataset. S1-S9 Figs dataset.**
(ZIP)

## Acknowledgments

We would like to Drs. E Tian and Lufeng Bai for offering technical assistance in ProliHH organoid studies, and Editage (www.editage.cn) for providing English editing services.

## Author Contributions

**Conceptualization:** Lin Yu, Hao Chang, Guoyu Pan, Ke Lan, Qiang Deng.

**Data curation:** Lin Yu, Hao Chang, Wentao Xie, Guoyu Pan, Ke Lan, Qiang Deng.

**Formal analysis:** Lin Yu, Hao Chang, Yuan Zheng, Guoyu Pan, Ke Lan, Qiang Deng.

**Funding acquisition:** Qiang Deng.

**Investigation:** Lin Yu, Hao Chang, Wentao Xie, Guoyu Pan, Ke Lan, Qiang Deng.

**Methodology:** Lin Yu, Hao Chang, Wentao Xie, Yuan Zheng, Le Yang, Qiong Wu, Fan Bu, Yuanfei Zhu.

**Project administration:** Guoyu Pan, Ke Lan, Qiang Deng.

**Resources:** Yuanfei Zhu, Youhua Xie, Guoyu Pan, Ke Lan, Qiang Deng.

**Supervision:** Guoyu Pan, Ke Lan, Qiang Deng.

**Validation:** Lin Yu, Hao Chang, Wentao Xie, Yuan Zheng, Le Yang, Qiong Wu, Fan Bu.

**Visualization:** Lin Yu, Hao Chang, Guoyu Pan, Ke Lan, Qiang Deng.

**Writing – original draft:** Lin Yu, Hao Chang, Qiang Deng.

**Writing – review & editing:** Lin Yu, Hao Chang, Wentao Xie, Yuan Zheng, Guoyu Pan, Ke Lan, Qiang Deng.

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
