## [Decision Letter · Decision Letter 0]

9 Sep 2024

Dear Dr. Deng,

Thank you very much for submitting your manuscript "Manganese is a potent inducer of lysosomal activity that inhibits de novo HBV infection" for consideration at PLOS Pathogens. As with all papers reviewed by the journal, your manuscript was reviewed by members of the editorial board and by several independent reviewers. In light of the reviews (below this email), we would like to invite the resubmission of a significantly-revised version that takes into account the reviewers' comments.

We cannot make any decision about publication until we have seen the revised manuscript and your response to the reviewers' comments. Your revised manuscript is also likely to be sent to reviewers for further evaluation.

Sincerely,

Alexander Ploss, Ph.D.

Academic Editor

PLOS Pathogens

Blossom Damania

Section Editor

PLOS Pathogens

Michael Malim

Editor-in-Chief

PLOS Pathogens

orcid.org/0000-0002-7699-2064

Reviewer's Responses to Questions

**Part I - Summary**

Reviewer #1: This is an interesting work on the late step of HBV entry process (endosomal escape/membrane fusion), which remains poorly understood following the discovery of NTCP as the functional HBV receptor more than a decade ago. First, the authors tested effect of manganese (Mn++) on HBV infectivity. Although Mn++ did not induce interferon response, it could still inhibit HBV infectivity in HepG2-NTCP cells. It was most effective if added before and/or during virus inoculation, and had little effect if added post-inoculation. Thus it affected the step of viral entry. Subsequent RNA-Seq followed by pathway analysis led them to test and validate mTORC1 as an Mn++ target and mediator of its inhibition of HBV infection. Thus reducing mTORC1 via serum deprivation or specific inhibitors (Torin1 and PP242) rather increased HBV infectivity in HepG2-NTCP cell line. Additional host factors in the early/late endosome and endosome/lysosome were examined. Chemicals that prevent acidification, including chloroquine and NH4Cl, improved HBV infectivity. Further studies confirmed that lysosomal activity is detrimental to HBV infectivity. Besides manganese, the authors also studied tomatidine, another activator of mTORC1. It moderately reduced HBV infectivity in both HepG2-NTCP cells and proliferating human hepatocytes (proliHH). In addition, the promoting effect of chloroquine and NH4Cl on HBV infectivity was also validated in proliHH. Overall, the studies were highly professional with complete data set, with the effect of mTORC1 demonstrated by both its stimulators (Mn++ and tomatidine) and inhibitors (FBS-free medium, Torin 1 and PP242). The work also shed light on why removing 10% FBS in the culture medium during HBV inoculation improves HBV infectivity.

Reviewer #2: Lin Yu and colleagues report in this manuscript that MnCl2 inhibits HBV infection of human hepatoma cells expressing NTCP via promotion of endocytosed HBV virions to be sorted into lysosomes for degradation. Mechanistically, the authors provide strong experimental evidence supporting the notion that MnCl2 treatment efficiently increases endocytic trafficking and endolysosomal acidification. The work significantly advances our understanding of the post-endocytosis events and regulation of HBV entry into hepatocytes.

Overall, the study is well conceived and executed. The conclusions are supported by the data presented. Further language editing should improve the manuscript.

My comments and suggestions are mostly minor and mainly for more clear/logical presentation and thorough discussion.

Reviewer #3: In this manuscript, Yu and colleagues serendipitously discovered that manganese (Mn²⁺) inhibits HBV infection in a cGAS/STING-independent but lysosomal activity-dependent manner. Using Mn²⁺ and its functional mimetics tomatidine as chemical probes, the authors further uncovered an ESCRT-mediated HBV post-entry sorting mechanism involving late endosomal escape. Along with this, activation of mTORC1 and subsequent endosome-lysosome fusion is detrimental to HBV infection, while inhibiting lysosome formation facilitates the process. Overall, this study provides novel and interesting insights into the cellular regulation of HBV post-entry trafficking and suggests that this mechanism could be exploited to develop new antiviral strategies for treating HBV infection. The presented results are largely compelling and supportive of the conclusions. The manuscript is also well-written together with well-organized figures. The following comments are provided for the authors to further strengthen the manuscript.

**Part II – Major Issues: Key Experiments Required for Acceptance**

Reviewer #1: Ref. 10 (Macovei A et al., J Virol 2013; 87: 6415-6427) also characterized the early steps of HBV entry process, but they used differentiated HepaRG cells rather than HepG2/NTCP cells. The current study validated finding from that report that RAB7 is essential for HBV infection (Fig. S5A). On the other hand, the previous work found treatment with 50mM NH4Cl did not affect HBV infectivity (they did not study chloroquine), whereas in this study, NH4Cl at 10mM or 5mM promoted HBV infection of HepG2-NTCP cells (Fig. 4E) or proliHH (Fig. 8E). Chloroquine had similar promotion of HBV infection. One possible explanation for the discordant findings is the different cell types used. Alternatively, the time point of treatment was different. In Ref. 10, NH4Cl was added 24hrs post-inoculation and the duration of treatment was 24hrs. In the present study, cells were pretreated with NH4Cl for 10hrs, followed by HBV inoculation for 12hrs (with or without NH4Cl? Not clearly indicated). What if the authors use the same schedule as the previous work (with NH4Cl added 24hrs post-inoculation rather than prior to inoculation)? If no promotion is observed, then the discordance could be attributed to different schedules.

Reviewer #2: 1. It will be interesting to experimentally demonstrate that MnCl2 and tomatidine treatment increase the degradation of internalized HBV protein and/or DNA in infected HepG2-NTCP cells.

Reviewer #3: 1. There is somewhat a disconnection between Figs.1-2 and Figs.3-4. In Fig.3, the authors demonstrated that mTOR suppression led to an increase in HBV-induced MVB formation. How was this connected to Mn2+ treatment? Why not directly treat with Mn2+ and then use an mTOR inhibitor as a control group? In Fig.4, VAMP7 was used to represent endosome-lysosome fusion. How is VAMP7 related to Mn2+ or mTOR? After siVAMP7 treatment, does Mn2+ still have an effect? Is mTOR activated? Some interesting information comes from the use of other inhibitors. Were these inhibitors applied continuously or only in the pre-infection stage? Furthermore, clarification is needed for the STX17-SNAP47-VAMP7/VAMP8 complex and its involvement in mTOR signaling.

2. While this study suggests a role of MVB/ESCRT in HBV post entry movement and cytosolic release, it is well-known that the MVB/ESCRT pathway is responsible for HBV virion secretion. Hence, the authors may want to check whether Mn2+ or tomatidine possesses any effect on HBV virion egress in their experimental system.

3. It would be ideal to graphically show the colocalization of HBV/NTCP complex and lysosomal marker(s) under Mn2+ treatment or mTORC1 activation.

**Part III – Minor Issues: Editorial and Data Presentation Modifications**

Reviewer #1: 1. The Discussion section may add something on limitations of work performed in vitro, mostly based on HepG2-NTCP cells. It will be interesting in the future to repeat the experiments of Mn++ and tomatidine treatment, VAMP7 knockdown, chloroquine and NH4Cl treatment, and bafilomycin A1 treatment in HepaRG cells or primary human hepatocytes. Moreover, all current in vitro HBV infection systems require artificial addition of 4% PEG, and a high genome copy number/cell. Thus one concern is whether the antiviral (therapeutic) effect of Mn++ and tomatidine observed in HepG2-NTCP cells can be observed in vivo (such as immunodeficient mice with humanized liver) under more authentic conditions of HBV infection. Under such conditions viral particles probably do not proceed to the endolysosome fraction.

2. Fig. 9 summarizes the major findings from this work and is supposed to be the highlight. But the current diagram is difficult to follow even with the figure legend. Some writings are extremely small, with some words even inverted. The word “Hepatocyte” at the bottom is not needed. What is “EV” on top right?

3. For supplementary figures, they should be called “Fig. S5” rather than “S5 Fig”.

Reviewer #2: 1. Line 41, because virus entry into host cells is generally considered to include all the molecular and cellular processes from receptor binding at cell surface to the delivery of viral capsids into the cytoplasm. Accordingly, endocytic trafficking of internalized HBV virion particles and endosomal/lysosomal escaping should still be considered as the part of viral entry process, but not “the molecular events of the viral post-entry steps”. More accurately, it should be post-endocytosis steps.

2. Line 47, put the sentence in a logic order, it should be “hyperactivation of mTORC1 and endo/lysosomal acidity”.

3. Line 130, a reference should be cited to support the claim that “Unlike many other enveloped viruses that employ low pH to facilitate membrane fusion, raising endolysosomal pH or inhibiting endosome-lysosome fusion promoted HBV infection.” If this statement is based on the findings reported in this manuscript, the paragraph should be rephrased to indicate this fact.

4. Logically, the data presented in Fig. 1 and Figs. S1 to S3 suggest MnCl2 treatment inhibits one or multiple steps of HBV infection post NTCP binding, but before viral RNA transcription. Therefore, the sentence in line 175 to 176 should be modified.

5. The data presented in Fig. 2C and D clearly indicate that MnCl2 treatment activates AKT-mTORC1 pathway in nutrient-rich condition. It will be very helpful to briefly interpret in the text why amino acid starvation completely abolishes the effects of MnCl2 activation of mTORC1 and downstream signaling pathway. Although not necessary, it will be very interesting to demonstrate or at least speculate/discuss how MnCl2 activates AKT.

6. It is very interesting that increased lysosomal acidification, but not activation of mTORC1, is more directly associated with the inhibition of HBV infection (Fig. S9). However, mTORC1 inhibitor efficiently promoted HBV infection (Fig. 3B). Why?

Reviewer #3: 1. Fig.1D-E, the figure labeling is unclear. Are the results represented in two replicates? Why was 100 μM not included? Was the treatment a one-time thing or repeated?

2. The activation of PI3K/AKT-mTOR has been shown to inhibit HBV replication by several previous studies, while this study did not observe direct antiviral effect on HBV replication upon Mn2+-induced AKT-mTOR activation (Fig. 2). Such discrepancy needs an explanation or discussion. The bottom panel of Fig. 2C shows a biphasic phosphorylation of AKT, any explanation?

3. Fig.S3, did the authors see a reduction in total NTCP protein in HBV-infected HepG2-NTCP cells upon Mn2+ treatment?

4. Fig.S4, what is the concentration of Mn in FBS? Does it matter?

5. Fig.8F, right panel, please include an uninfected control. Do the authors have additional data to support a productive HBV infection in the ProliHH organoids such as intracellular HBV RNA and DNA?

6. While Mn2+ did not inhibit HBV DNA replication in HepAD38 cells (Fig. S2B), tomatidine treatment resulted in an obvious reduction of HBV DNA in the same cells (Fig. S8A). The authors may discuss the possible reason for such discrepancy. Is it related to the different degrees of mTORC1 activation between these two chemicals?

7. Line 106, “primary” should be “primarily”.

PLOS authors have the option to publish the peer review history of their article (what does this mean?). If published, this will include your full peer review and any attached files.

Reviewer #1: No

Reviewer #2: No

Reviewer #3: No
---

## [Decision Letter · Decision Letter 1]

2 Dec 2024

Dear Dr. Deng,

We are pleased to inform you that your manuscript 'Manganese is a potent inducer of lysosomal activity that inhibits de novo HBV infection' has been provisionally accepted for publication in PLOS Pathogens.

Best regards,

Alexander Ploss, Ph.D.

Academic Editor

PLOS Pathogens

Blossom Damania

Section Editor

PLOS Pathogens

Michael Malim

Editor-in-Chief

PLOS Pathogens

orcid.org/0000-0002-7699-2064

Reviewer Comments (if any, and for reference):

Reviewer's Responses to Questions

**Part I - Summary**

Reviewer #1: (No Response)

Reviewer #2: The authors have thoroughly addressed my concerns on the previous version with complete satisfaction.

Reviewer #3: The authors have adequately addressed my previous comments.

**Part II – Major Issues: Key Experiments Required for Acceptance**

Reviewer #1: (No Response)

Reviewer #2: No.

Reviewer #3: (No Response)

**Part III – Minor Issues: Editorial and Data Presentation Modifications**

Reviewer #1: (No Response)

Reviewer #2: No.

Reviewer #3: (No Response)

PLOS authors have the option to publish the peer review history of their article (what does this mean?). If published, this will include your full peer review and any attached files.

Reviewer #1: No

Reviewer #2: No

Reviewer #3: No

---

## [Editor Report · Acceptance letter]

11 Dec 2024

Dear Dr. Deng,

We are delighted to inform you that your manuscript, "Manganese is a potent inducer of lysosomal activity that inhibits de novo HBV infection," has been formally accepted for publication in PLOS Pathogens.

Best regards,

Sumita Bhaduri-McIntosh

Editor-in-Chief

PLOS Pathogens

orcid.org/0000-0003-2946-9497

Michael Malim

Editor-in-Chief

PLOS Pathogens

orcid.org/0000-0002-7699-2064